# Subgraph Permutation Equivariant Networks

**Joshua Mitton**                                                                *j.mitton.1@research.gla.ac.uk*
*School of Computing Science*
*University of Glasgow, Glasgow, UK*

**Roderick Murray-Smith**                                          *roderick.murray-smith@glasgow.ac.uk*
*School of Computing Science*
*University of Glasgow, Glasgow, UK*

**Reviewed on OpenReview:** *https://openreview.net/forum?id=5erasT6Tal*

## Abstract

In this work we develop a new method, named Sub-graph Permutation Equivariant Networks (SPEN), which provides a framework for building graph neural networks that operate on sub-graphs, while using a base update function that is permutation equivariant, that are equivariant to a novel choice of automorphism group. Message passing neural networks have been shown to be limited in their expressive power and recent approaches to over come this either lack scalability or require structural information to be encoded into the feature space. The general framework presented here overcomes the scalability issues associated with global permutation equivariance by operating more locally on sub-graphs. In addition, through operating on sub-graphs the expressive power of higher-dimensional global permutation equivariant networks is improved; this is due to fact that two non-distinguishable graphs often contain distinguishable sub-graphs. Furthermore, the proposed framework only requires a choice of $k$-hops for creating ego-network sub-graphs and a choice of representation space to be used for each layer, which makes the method easily applicable across a range of graph based domains. We experimentally validate the method on a range of graph benchmark classification tasks, demonstrating statistically indistinguishable results from the state-of-the-art on six out of seven benchmarks. Further, we demonstrate that the use of local update functions offers a significant improvement in GPU memory over global methods.

## 1 Introduction

Machine learning on graphs has received much interest in recent years with many graph neural network (GNN) architectures being proposed. One such method, which is widely used, is the general framework of message passing neural networks (MPNN). These provide both a useful inductive bias and scalability across a range of domains (Gilmer et al., 2017).

However, Xu et al. (2019); Morris et al. (2019b) showed that models based on a message passing framework with permutation invariant aggregation functions have expressive power at most that of the Weisfeiler-Lehman (WL) graph isomorphism test (Weisfeiler & Leman, 1968). Therefore, there exist many non-isomorphic graphs that a model of this form cannot distinguish between. Figure 1 provides an example of two non-isomorphic graphs which to a message passing update function are indistinguishable.

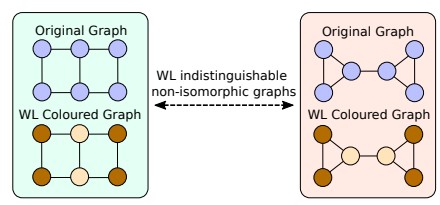

Figure 1: The initial graph on the left is non-isomorphic to the graph on the right. Despite this the WL graph isomorphism test cannot distinguish between the two graphs. Adapted from Bevilacqua et al. (2021).

This presents a natural question of is it possible to design a GNN that improves the expressive power of MPNNs? Many methods have been proposed to address this question, but most often an increase in expressivity must be traded off against scalability. We present the background into existing methods which attempt to tackle this question in Section 2.

**Our approach**. We design a framework to create provably more expressive and scalable graph networks. We achieve this through incorporating symmetry structures in graphs, by considering a graph equivariant update function which operates over sub-graphs. Our framework, dubbed Subgraph Permutation Equivariant Networks (SPEN), is developed from the observation that operating on sub-graphs both improves the scalability and expressive power of higher-dimensional GNNs, whilst unlocking a natural choice of automorphism groups which further increases the expressive power of the network. Our framework consists of:

1. encoding the graph as a bag of bags of sub-graphs,
2. utilising a $k$-order permutation equivariant base encoder, and
3. constraining the linear map to be equivariant to the automorphism groups of the bags of sub-graphs.

Sub-graphs each have a symmetry group and our framework captures this in two ways. Each sub-graph has a permutation symmetry, which is induced by a permutation of the nodes in the graph. In addition, there is a symmetry across sub-graphs whereby sub-graphs are associated to an automorphism group. We therefore construct a neural network comprising of layers that are equivariant to both permutations of nodes and the automorphism groups of sub-graphs. We achieve this by utilising a permutation equivariant base encoder with feature space constrained by the direct sum of different order permutation representations. Further, we constrain the linear map comprising each layer to be equivariant to the automorphism groups of the bags of sub-graphs. This necessitates that sub-graphs belonging to different automorphism groups are processed by a kernel with different weights, while for sub-graphs belonging to the same automorphism group the kernel shares weights. This leads to us creating a sub-graph extraction policy which generates a bag of bags of sub-graphs, where each bag of sub-graphs corresponds to a different sub-graph automorphism group.

**Main contributions**

1. An automorphism equivariant compatible subgraph extraction method.
2. A novel choice of automorphism groups with which to constrain the linear map to be equivariant to.
3. A more scalable framework for utilising higher-dimensional permutation equivariant GNNs.
4. A more expressive model than higher-dimensional permutation equivariant GNNs and sub-graph MPNNs.
5. A theoretical analysis of the proposed model in terms of scalability and an application of the theoretical analysis from Bevilacqua et al. (2021) and de Haan et al. (2020) to demonstrate the expressivity of our model.
6. A demonstration that our method is statistically indistinguishable from state-of-the-art methods on benchmark graph classification tasks.

Throughout this paper we are following the notation presented by (Bevilacqua et al., 2021) for how we present what a subgraph is and for the expressivity analysis.[1]

## 2 Background

More expressive graph neural networks (GNNs) exist which can be grouped into three different groups: (1) those which design higher-dimensional GNNs, (2) those which use positional encodings through pre-coloring nodes, and (3) those which use sub-graphs/local equivariance. Several architectures have been proposed of the type (1) which design a high-order GNN equivalent to the hierarchy of $k$-WL tests (Maron et al., 2018; 2019; Morris et al., 2019b;a; Keriven & Peyré, 2019; Azizian & Lelarge, 2021; Zhang & Li, 2021). Despite being equivalent to the $k$-WL test, and hence having provably strong expressivity; these models lose the advantage of locality and linear complexity. As such, the scalability of such models poses an issue for their practical use, with Maron et al. (2018) showing that the basis space for permutation equivariant models of order $k$

---

[1]One of contributions of our paper namely operating on subgraphs, which inherit ids from the original graph was developed concurrently to (Bevilacqua et al., 2021), see `https://openreview.net/forum?id=7oyVOECcrt` for the initial version of our work.

is equal to the $2k^{th}$ Bell number, which results in a basis space of size 2 for order-1 tensors, 15 for order-2 tensors, 203 for order-3 tensors, and 4140 for order-4 tensors, demonstrating the practical challenge of using higher-dimensional GNNs. Several architectures have also been proposed of type (2) where authors seek to introduce a pre-coloring or positional encoding that is permutation invariant. These comprise of pre-coloring nodes based on pre-defined substructures (Bouritsas et al., 2020) or lifting graphs into simplicial- (Bodnar et al., 2021b) or cell complexes (Bodnar et al., 2021a). These methods require a pre-computation stage, which in the worst-case finding substructures of size $k$ in a graph of $n$ nodes is $\mathcal{O}(n^k)$. Finally sub-graphs/local equivariance of type (3) have been considered to find more expressive GNNs. Local graph equivariance requires a (linear) map that satisfies an automorphism equivariance constraint. This is due to the nature of graphs having different local symmetries on different nodes/edges. This has been considered by de Haan et al. (2020) though imposing an isomorphism/automorphism constraint on edge neighbourhoods and by Thiede et al. (2021) by selecting specific automorphism groups and lifting the graph to these. Although the choice of automorphism group chosen by de Haan et al. (2020) leads to little weight sharing and requires the automorphism constraint to be parameterized, while those proposed by Thiede et al. (2021) and Xu et al. (2021) do not guarantee to capture the entire graph and requires a hard-coded choice of automorphism group. Xu et al. (2021) also propose a method for searching across different sub-graph templates, although this still requires some hard-coding. Operating on sub-graphs has been considered as a means to improve GNNs by dropping nodes (Papp et al., 2021; Cotta et al., 2021), dropping edges (Rong et al., 2019), utilising ego-network graphs (Zhao et al., 2021), and considering the symmetry of a bag of sub-graphs (Bevilacqua et al., 2021).

## 3   Subgraph Permutation Equivariant Networks (SPEN)

We first outline necessary definitions that our method builds upon. Following this we present the two main concepts of our SPEN model. Firstly, SPEN has a $k$-ego net subgraph extraction method, which are collected into bags determined by their automorphism group. Secondly, SPEN has an automorphism equivariant graph neural network. This section presents the core concepts of the model which contribute to the improved expressivity and classification performance. The overall architecture of the SPEN model is presented in Figure 2. Further, the breakdown of an automorphism equivariant layer within our framework is presented in Figure 3. In addition, in Figure 4 we detail an example breakdown of one of the mapping functions used within our model. Finally, Figure 5 shows a visualisation of some of the bases used within the mapping functions in the automorphism equivariant functions. In addition, we present a more general overview of the architectural details of the SPEN framework in Appendix A.5. The key definitions required to understand our framework are provided below, with further useful definitions provided in Appendix A.1.

### 3.1   Definitions

In this work we consider graphs as concrete graphs and utilise sub-concrete graphs in our framework. The sub-graphs are extracted as $k$-ego network sub-graphs. This leads us to define the graphs and sub-graphs.

**Definition 3.1.** A *Concrete Graph* (de Haan et al., 2020) $G$ is a finite set of nodes $\mathcal{V}(G) \subset \mathbb{N}$ and a set of edges $\mathcal{E}(G) \subset \mathcal{V}(G) \times \mathcal{V}(G)$.

The set of node ids may be non-contiguous and we make use of this here as we extract overlapping sub-graphs and perform the graph update function on this bag of sub-graphs. "The natural numbers of the nodes are essential for representing the graphs in a computer, but hold no actual information about the underlying graph" (de Haan et al., 2020). Therefore, the same underlying graph can be given in may forms by a permutation of the ordering of the natural numbers of the nodes. Throughout the paper we refer to concrete graphs as graphs to minimise notation.

**Definition 3.2.** In tensor format the values of $G$ are encoded in a tensor $\mathbf{A} \in \mathbb{R}^{|\mathcal{V}(G)| \times |\mathcal{V}(G)| \times d}$.

The node features are encoded along the diagonal and edge features encoded in off-diagonal positions.

**Definition 3.3.** A *sub-Concrete Graph H* is created by taking a node $i \in \mathcal{V}(G)$, and extracting the nodes $j \in \mathcal{V}(G)$ and edges $(i, j) \subset \mathcal{V}(G) \times \mathcal{V}(G)$, according to some sub-graph selection policy.

In this work we consider the sub-graph selection policy as a *k*-ego-network policy. For brevity we refer to sub-concrete graphs as sub-graphs throughout the paper.

**Definition 3.4.** A *k-Ego Network* of a node is its *k*-hop neighbourhood with induced connectivity.

In this work we are interested in the symmetries between graphs. For this we considered the automorphism symmetries of graphs through consideration of the automorphism group. Here we consider the automorphism group of the permutation group and therefore make some necessary definitions for the consideration of groups and their representations.

**Definition 3.5.** A *group representation* $\rho$ of the group $G$ is a homomorphism $\rho : G \to \mathrm{GL}(V)$ of $G$ to the group of automorphisms of $V$ (Fulton & Harris, 2013). A group representation associates to each $g \in G$ an invertible matrix $\rho(g) \in \mathbb{R}^{n \times n}$. This can be understood as specifying how the group acts as a transformation on the input.

**Definition 3.6.** A *feature space* is a vector space $V$ with a group representation $\rho$ acting on it. The choice of group representations on the input and output vector spaces of a linear map constrains the possible forms the linear map can take.

**Definition 3.7.** A *tensor representation* can be built up from some base group representations $\rho(g)$ through the tensor operations dual ($*$), direct sum ($\oplus$), and tensor product ($\otimes$). This allows for tensor representations to be constructed that are of increasing size and complexity.

**Definition 3.8.** A *kernel constraint* is taken to mean a restriction of the space a kernel or linear map can take between two vector spaces. The symmetric subspace of the representation is the space of solutions to the constraint $\forall g \in G : \rho(g)v = v$, which provides the space of permissible kernels.

As was said previously, we are interested in the symmetries between graphs. Here we explore the automorphism symmetries of graphs and hence define what an automorphism group is. Further, we define a naturality constraint for a linear map as this governs a symmetry condition up to graph isomorphism.

**Definition 3.9.** Automorphism groups.
Let $\mathcal{X}$ be some mathematical object for which we can formulate the notion of homomorphism (or isomorphism). Then an automorphism of $\mathcal{X}$ is an isomorphism $\theta : \mathcal{X} \to \mathcal{X}$; in other words, it is a permutation of $\mathcal{X}$ which happens also to be a homomorphism satisfying

$$(x \circ y)\theta = x\theta \circ y\theta.$$

Let $\mathrm{Aut}(\mathcal{X})$ be the set of all automorphisms of $\mathcal{X}$. Then $\mathrm{Aut}(\mathcal{X})$ is a group, the automorphism group of $\mathcal{X}$. This can also be considered for a group rather than a general object $\mathcal{X}$ and therefore we can talk about the automorphism group of a group.

**Definition 3.10.** A *Graph isomorphism*, $\phi : G \to G'$ is a bijection between the vertex sets of two graphs $G$ and $G'$, such that two vertices $u$ and $v$ are adjacent in $G$ if and only if $\phi(u)$ and $\phi(v)$ are adjacent in $G'$. This mapping is edge preserving, i.e. satisfies for all $(i, j) \in \mathcal{V}(G) \times \mathcal{V}(G)$:

$$(i, j) \in \mathcal{E}(G) \iff (\phi(i), \phi(j)) \in \mathcal{E}(G').$$

An isomorphism from the graph to itself is known as an automorphism.

**Definition 3.11.** The *naturality* constraint for a linear map states that for a graph $G$ and linear map $f_G : \rho(G) \to \rho'(G)$ the following condition holds for every graph isomorphism $\phi$:

$$\rho'(\phi) \circ f_G = f_{G'} \circ \rho(\phi).$$

Following the definitions it is noteworthy that when considering the permutation symmetries of a graph and looking at the automorphism group, there can be no automorphic mapping between a graph with four nodes and a graph with five nodes. This is due to there being no permutation of the four nodes features such that they yield the five nodes features.

In this paper we are interested in the symmetries of the symmetric group $S_n$. This constraint can be solved for different order tensor representations (Maron et al., 2018; Finzi et al., 2021). We present the space of linear layers mapping from $k$-order representations to $k'$-order representations in Figure 5.

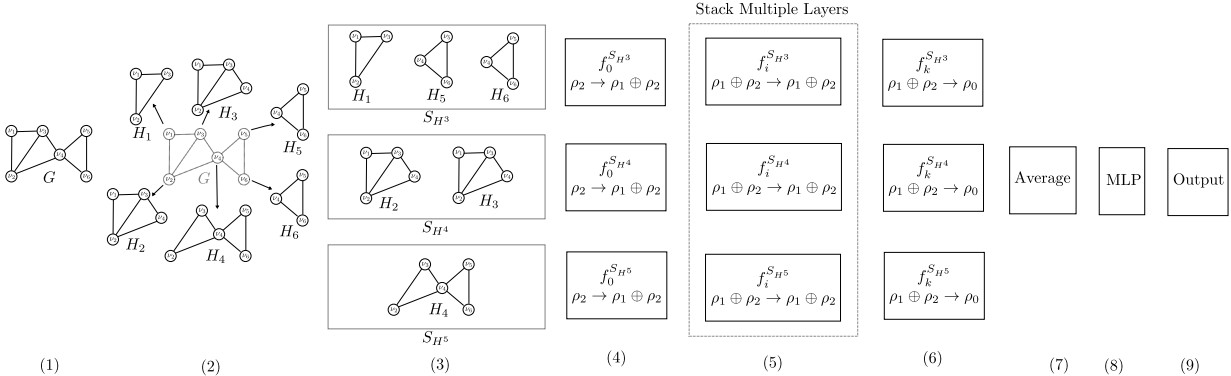

Figure 2: (1-2) Splitting the graph into sub-graphs. (3) Place sub-graphs into bags , where each bag holds sub-graphs of a specific size. (4) Process the bags of sub-graphs with an automorphism equivariant linear map. (5) Stack multiple layers each comprising of an automorphism equivariant mapping function. (6) Add a final automorphism equivariant mapping function. (7) Each automorphism group is averaged. (8) An MLP is used to update the feature space.

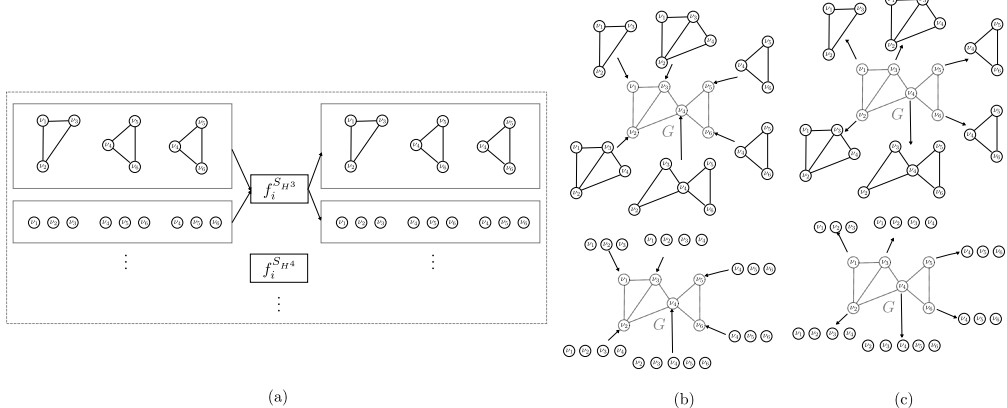

Figure 3: This figure breaks down what a single automorphism equivariant layer within our model looks like. In Figure 2 this corresponds to looking inside a single dashed box in (5). Here we see in (a) that the input to the layer is a vector space transforming under the group representation $\rho_1 \oplus \rho_2$ corresponding to graphs and sets as inputs. These processed by automorphism equivariant update functions $f$, where there is an $f^{S_{H^i}}$ for each automorphism group. (b) Following the automorphism equivariant update function we re-insert the vector space features back into their respective nodes in the original graph, and (c) re-extract back into the respective sub-graphs. This can be seen as a form of narrowing and promotion and allows information to propagate between sub-graphs.

## 3.2 Sub-graph Selection Policy

Sub-graphs can be extracted from a graph in a number of ways, by removing nodes, by removing edges, extracting connectivity graphs at nodes, or extracting connectivity graphs at edges to name a few. In this work we focus on $k$-ego network sub-graphs. These are sub-graphs extracted by considering the $k$-hop connectivity of the graph at a selected node and extracting the induced connectivity. The sub-graph selection policy of $k$-ego networks therefore extracts a sub-graph for each node in the original graph.

In this work we process graphs as bags of sub-graphs. In general the size of the sub-graphs, $|H_n|$, extracted for a graph are not all the same size, and thus $|H_n|$ varies from sub-graph to sub-graph. We therefore go further than representing each graph as a bag of sub-graphs and represent each graph as a bag of bags of sub-graphs, where each bag of sub-graphs hold sub-graphs of the same size, i.e. $S_{H^i}$ is the bag of sub-graphs for sub-graphs of size $|H_n| = i$. The graph is therefore represented as the bag of bags of sub-graphs

$G \equiv \{\{S_{H^i} \dots S_{H^k}\}\} \equiv \{\{\{\{H_1^i, ..., H_a^i\}\}, ..., \{\{H_1^k, ..., H_c^k\}\}\}\}$, for sub-graphs $H$, with bags of sub-graphs which are each of size $a, ..., c$ containing sub-graphs of sizes $i, .., k$ respectively.

In Chapter 4 we demonstrate how our choice of sub-graph selection policy improves the expressivity of the overall model. Further, in Chapter 4 and 5 we show that the choice of sub-graph selection allows the method to scale better than global approaches. Finally, in Chapter A.6 we show that using $k$-ego network sub-graphs yields sub-graphs which are typically much smaller than the original graph. This both feeds into the improved ability of our method to scale to larger graphs and provides a smaller more compact automorphism group compared to an approach such as node removed sub-graphs. As a result our method requires less parameterisation of the automorphism group than other automorphism equivariant approaches.

### 3.3 Automorphism Equivariant Graph Network Architecture

The input data represented as a bag of bags of sub-graphs has a symmetry group of both the individual sub-graphs and of the bags of sub-graphs. We construct a graph neural network that is equivariant to this symmetry. This can be broken down into three parts: (1) the automorphism symmetry of the bags of sub-graphs, (2) the permutation symmetry of sub-graphs within bags of sub-graphs and the original graph permutation symmetry, (3) sub-graph linear maps.

An overview of the architecture is detailed in Figure 2, where each component is described: (1-2) The first component of our SPEN model comprises of splitting the graph $G$ into sub-graphs $H_1 \dots H_n$, for a graph with $n$ nodes. For this we use a $k$-ego network policy extracting a sub-graph for each node in the input graph. (3) Secondly, we place sub-graphs $H_1 \dots H_n$ into bags $S_{H^i} \dots S_{H^j}$, where each bag holds sub-graphs of a specific size, with $i \dots j$ being the different sizes $|H|$ of sub-graphs. The extracted sub-graphs are used as fully-connected graphs with zero features for non-edges; this results in each bag of sub-graphs representing an automorphism group. Here it is worth noting that the figure shows three bags of sub-graphs or three automorphism groups, while in general there does not have to just be three automorphism groups and this can vary. (4) We then process the bags of sub-graphs with an automorphism equivariant linear map. This comprises of multiple separate functions $f$, with a different function processing each automorphism group, i.e. $f_0^{S_{H^3}}$ is the function mapping the automorphism group corresponding to the bag $S_{H^3}$ of sub-graphs in layer 0. This is a map from a 2-order permutation representation, i.e. graphs, to the direct sum of a 1-order and 2-order permutation representation. (5) We then stack multiple layers each comprising of an automorphism equivariant mapping function. Each of the automorphism groups is updated with a function mapping from the direct sum of a 1-order and 2-order permutation representation to the direct sum of a 1-order and 2-order permutation representation. (6) The final layer in the model is again an automorphism equivariant mapping function were each automorphism group is mapped from the direct sum of a 1-order and 2-order permutation representation to a 0-order representation. (7) Each automorphism group is averaged. (8) An MLP is used to update the feature space.

#### 3.3.1 Automorphism Symmetry

We have defined the sub-graph selection policy used, which results in the input graph, $G$, being represented as a bag of bags of sub-graphs, $\{\{S_{H^i} \dots S_{H^k}\}\}$. As each bag of sub-graphs holds sub-graphs of a different size, each forms a different automorphism group. Therefore, we have different feature spaces for different sub-graph sizes, i.e. $\rho(S_{H^i}) \neq \rho(S_{H^j})$. A linear layer acting on sub-graphs should therefore operate differently on sub-graphs from different automorphism groups. This is demonstrated in Figure 2 (4,5,6) by having different linear map $f$ for each bag of sub-graphs ($f_i^{S_{H^3}}, f_i^{S_{H^4}}, f_i^{S_{H^5}}$). This is defined more rigorously through the concept of naturality. Given a linear layer mapping from a feature space acted upon by $\rho_m$ to a feature space acted upon by $\rho'_m$ for each sub-graph $H$ a (linear) map can be detailed as $f_H : \rho_m(H) \rightarrow \rho'_m(H)$. However, given two isomorphic sub-graphs $H$ and $H'$ are the same graph up-to some bijective mapping, we want $f_H$ and $f_{H'}$ to process the feature spaces in an equivalent manner. This naturality constraint is therefore similar to the global naturality constraint on graphs $G$, for sub-graphs, $H$:

$$\rho'(\phi) \circ f_H = f_{H'} \circ \rho(\phi). \tag{1}$$

This constraint (Equation 1) says that if we first transition from the input feature space acted upon by $\rho(H)$ to the equivalent input feature space acted upon by $\rho'(H)$ via an isomorphism transformation $\rho(\phi)$ and then apply $f_{H'}$ we get the same thing as first applying $f_H$ and then transitioning from the output feature space acted upon by $\rho'(H)$ to $\rho'(H')$ via the isomorphism transformation $\rho'(\phi)$. Since $\rho(\phi)$ is invertible, if we choose $f_H$ for some $H$ then we have determined $f_{H'}$ for any isomorphic $H'$ by $f_{H'} = \rho'(\phi) \circ f_H \circ \rho(\phi)^{-1}$. Therefore, for any automorphism $\phi: H \to H$, we get an equivariance constraint $\rho'(\phi) \circ f_H = f_H \circ \rho(\phi)$. Thus, a layer in the model must have for each automorphism group a map $f_H$ that is equivariant to automorphisms. Therefore our choice of sub-graph selection policy, extracting bags of sub-graphs, aligns with the naturality constraint, in that we require a mapping function $f^{S_{H^i}}$ for each bag of sub-graphs.

### 3.3.2 Permutation Symmetries Within Bags of Sub-graphs

The order of sub-graphs in each bag of sub-graphs is arbitrary and changes if the input graph is permuted. This ordering comes from the need to represent the graph in a computer and the ordering is tracked through the use of concrete graphs. It would therefore be undesirable for the output prediction to be dependent upon this arbitrary ordering. This is overcome in the choice of insert and extract functions used to share information between sub-graphs demonstrated in Figure 3. At the end of a linear layer in our model node and edge features from the original graph can be represented multiple times, i.e. it occurs in multiple sub-graphs. We therefore average these features across sub-graphs through and insert and extract function and in doing this ensure the output is invariant to the ordering of sub-graphs in each bag.

### 3.3.3 Sub-graph Linear Maps

Each sub-graph has a symmetry group that is given by permutation of the order of nodes in the graph. This group is denoted $S_n$ for a graph of $n$ nodes. The group $S_n$ acts on on the graph via $(\sigma \cdot A)_{ij} = A_{\sigma^{-1}(i)\sigma^{-1}(j)}$. Sub-graphs, $H$, therefore have a symmetry group $S_m \le S_n$ and we are interested in constructing graph neural network layers equivariant to this symmetry group. The graph is an order-2 tensor and the action of the permutation group can be generalised to differing order tensors. For example, the set of nodes in a graph is an order-1 tensor. For the case of a linear mapping from order-2 permutation representations to order-2 permutation representations, the basis space was shown to comprise of 15 elements by Maron et al. (2018). Similarly, the constraint imposed by equivariance to the group of permutations can be solved for different order representation spaces and we provide an example of all mappings between representation spaces from order 0-2 in Figure 5. We are not restricted to selecting a single input-output order permutation representation space and can construct permutation equivariant linear maps between multiple representations separately through the direct

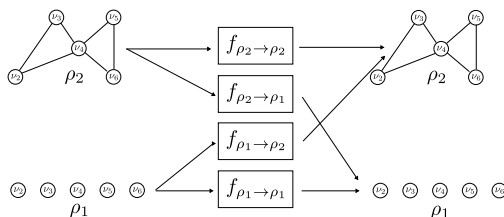

Figure 4: An example of what a function box in Figure 2 breaks down into. This example is for a function $f_i^{S_H}$ mapping from a representation $\rho_1 \oplus \rho_2 \to \rho_1 \oplus \rho_2$. This demonstrates there is a mapping function from $\rho_2$ to $\rho_2$, from $\rho_2$ to $\rho_1$, from $\rho_1$ to $\rho_2$, and from $\rho_1$ to $\rho_1$, as well as $\rho_2$ is a group representation for graphs and $\rho_1$ is a group representations for sets.

sum $\oplus$. For example the direct sum of order 1 and 2 representations is given by $\rho_1 \oplus \rho_2$. We utilise this to build the linear mapping functions, $f^{S_{H^i}}$, shown in Figures 2 (4,5,6) and 3 (a) to use linear maps between feature space acted on by different representations $\rho$. An example of how this map breaks down into individual functions mapping from and to a graph feature space acted on by 2-order permutation representations and a set feature space acted on by 1-order permutation representations is provided in Figure 4. Further, what the bases utilised within each of these functions mapping between feature spaces acted upon by different order representations is given in Figure 5.

Due to the construction of a sub-graph the sub-graphs inherit node ids from the original graph. Therefore, a permutation of the order of the nodes in the original graph corresponds to an equivalent permutation of the ordering of the nodes in the sub-graphs. In addition, as the permutation action on the graph does not change the underlying connectivity, the sub-graphs exacted are individually unchanged up-to some isomorphism. Therefore, a permutation of the graph only permutes the ordering at which the sub-graphs are extracted.

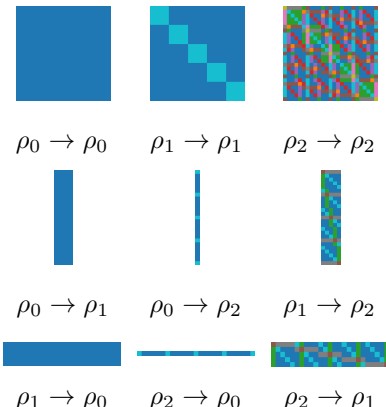

Figure 5: Bases for mappings to and from different order permutation representations, where $\rho_k$ is a $k$-order representation. Each color in a basis indicates a different parameter. $\rho_2 \to \rho_2$ is a mapping from a graph to a graph, and has 15 learnable parameters. Further, there are mappings between different order representation spaces and higher order representation spaces.

### 3.4 Related Work

We have largely discussed the related methods to our work in Section 2. Despite this, we provide a more extensive explanation of some other methods in Appendix A.3 and demonstrate how some of these methods can be implemented within our framework in Appendix A.4.

## 4 Analysis of Expressivity and Scalability

In this section we study both the expressive power of our architecture following the theoretical analysis outlined in Bevilacqua et al. (2021) by its ability to provably separate non-isomorphic graphs and the scalability by its ability to process larger graphs that its predecessor.

### 4.1 WL Test and Expressive Power

The Weisfeiler-Lehman (WL) test (Weisfeiler & Leman, 1968) is a graph isomorphism test commonly used as a measure of expressivity in GNNs. This is due to the similarity between the iterative color refinement of the WL test and the message passing layers of a GNN. The WL test is a necessary but insufficient condition, which is not able to distinguish between all non-isomorphic graphs. The WL test was extended to the $k$-WL test, which provides increasingly more powerful tests that operate on $k$-tuples of nodes.

**WL analogue for sub-graphs.** One component of our model is the idea of operating on sub-graphs rather than the entire graph, more specifically our architecture operates on ego-network sub-graphs. We follow the theoretical analysis outlined in Bevilacqua et al. (2021) to verify that operating on sub-graphs will improve the expressive power of the base model within our automorphism equivariant model. Therefore, following Bevilacqua et al. (2021) we present a color-refinement variant of the WL isomorphism test that operates on a bag of sub-graphs.

**Definition 4.1.** The sub-graph-WL test utilises a color refinement of $c_{v,S}^{t+1} = \text{HASH}(c_{v,S}^t, \mathcal{N}_{v,S}^t, C_v^t)$, which is a simplification of that outlined in (Bevilacqua et al., 2021), where $\text{HASH}(\cdot)$ is an injective function, $\mathcal{N}_{v,S}^t$ is the node neighbourhood of $v$ within the ego-network sub-graph $S$, and $C_v^t$ is the multiset of $v$'s colors across sub-graphs.

**Theorem 4.2.** *Sub-graph-WL is strictly more powerful than 1&2-WL.*

In Appendix A.2 we prove Theorem 4.2. This yields the result that even for a simple 1-WL expressive function in the GNN, such as message passing, the model is immediately more expressive than 1&2-WL.

**Comparing SPEN to the WL test**. We have already shown that when considering a graph update function that operates on a bag of $k$-ego network sub-graphs, even if the update function itself has limited

expressivity, it is more expressive than 1&2-WL. SPEN utilises a natural permutation equivariant update function through operating on a bag of bags of sub-graphs. The naturality constraint of our model states that each automorphism group of sub-graphs should be processed by a different (linear) map. In addition, we utilise higher-dimensional GNNs. Both of these choices are expected to increase the expressive power of our model.

**Proposition 4.3.** *For two non-isomorphic graphs $G^1$ and $G^2$ sub-graph-WL can successfully distinguish them if (1) they can be distinguished as non-isomorphic from the multisets of sub-graphs and (2)* $\text{HASH}(\cdot)$ *is discriminative enough that* $\text{HASH}(c_{v,S^1}^t, \mathcal{N}_{v,S^1}^t, C_v^t) \neq \text{HASH}(c_{v,S^2}^t, \mathcal{N}_{v,S^2}^t, C_v^t)$ .

This implies that despite the sub-graph policy increasing the expressive power of the model, it is still limited by the ability of the equivalent to the $\text{HASH}(\cdot)$ function's ability to discriminate between the bags of sub-graphs. The naturality constraint of our model processing each automorphism group with a different higher-dimensional GNN is therefore expected to increase the expressive power of our model over sub-graph methods utilising a MPNN.

**Theorem 4.4.** *SPEN is strictly more powerful than sub-graph MPNN.*

We demonstrate the claim of Theorem 4.4 similarly to Bouritsas et al. (2020); de Haan et al. (2020). We use a neural network with random weights on a graph and compute a graph embedding. We say the neural network finds two graphs to be different if the graph embeddings differ by an $\ell_2$ norm of more than $\epsilon = 10^{-3}$ of the mean $\ell_2$ norms of the embeddings of the graphs in the set. The network is said to be most expressive if it only finds non-isomorphic graphs to be different. We test this by considering a set of 100 random non-isomorphic non-regular graphs, a set of 100 non-isomorphic graphs, a set of 15 non-isomorphic strongly regular graphs,[2] and a set of 100 isomorphic graphs. Table 1 shows that a simple invariant message passing (GCN) (Kipf & Welling, 2016) as well as a simple invariant message passing model operating on sub-graphs (SGCN), which we created, are unable to distinguish between regular and strongly regular graphs. Further, it is shown that PPGN (Maron et al., 2019) can distinguish regular graphs but not strongly regular graphs, although a variant of PPGN that uses high order tensors should also be able to distinguish strongly regular graphs. On the other hand, our SPEN model is able to distinguish strongly regular graphs. Therefore, our model is able to distinguish non-isomorphic graphs that a sub-graph MPNN cannot and is strictly more powerful.

Table 1: Rate of pairs of graphs in the set of graphs found to be dissimilar in expressiveness experiment. An ideal method only find isomorphic graphs dissimilar. A score of 1 implies the model can find all graphs dissimilar, while 0 implies the model finds no graphs dissimilar.

| Model | Random | Regular | Str. Regular | Isom. |
|-------|--------|---------|--------------|-------|
| GCN   | 1      | 0       | 0            | 0     |
| SGCN  | 1      | 0       | 0            | 0     |
| PPGN  | 1      | 0.97    | 0            | 0     |
| SPEN  | 1      | 0.98    | 0.97         | 0     |

## 4.2 Scalability

Global permutation equivariant models of the form found by Maron et al. (2018) operate over the entire graph. They therefore scale with $\mathcal{O}(n^2)$, for graphs with $n$ nodes. Our method operates on $k$-ego network sub-graphs where a sub-graph is produced for each node in the original graph. Our method therefore scales with $\mathcal{O}(nm^2)$, where $m$ is the number of nodes in the $k$-ego network sub-graph. It is therefore clear that if $n = m$, theoretically, our method scales more poorly than global permutation equivariant models, although this would imply the graph is fully-connected and every sub-graph is identical. In this situation extracting sub-graphs is irrelevant and only 1 sub-graph is required (the entire graph) and hence if $n = m$ our method scales with that of global permutation equivariant models. The more interesting situation, which forms

---

[2]See `http://users.cecs.anu.edu.au/~bdm/data/graphs.html`.

the majority of graphs, is when $n \neq m$. When $m \ll n$ our method scales more closely with methods that scale linearly with the size of the graph and it is for this type of data that our method offers a significant improvement in scalability over global permutation equivariant models.

We empirically show how SPEN and global permutation equivariant methods scale depending on the size of $n$ and $m$ by analysing the GPU memory usage of both models across a range of random regular graphs. We utilise random regular graphs for the scalability test as it allows for precise control over the size of the overall graph and sub-graphs. We compare the GPU memory usage of both models across a range of graph sizes with a sub-graph size of $m = 3, 6,$ and 9. Through analysing the graphs in the TUDataset, which we make use of when experimenting on graph benchmarks, we note that the average sub-graph sizes range between 3 and 10 (see Table 3), justifying the choice of sub-graphs in the scalability tests. Figure 6 shows that the Global Permutation Equivariant Network (GPEN) (Maron et al., 2018) cannot scale beyond graphs with 500 nodes. On the other hand, our method (SPEN) scales to larger graphs of over an order of magnitude larger. In the situation where $m = 3$ GPEN can process graphs of size up to 500 nodes, while our SPEN can process graphs of size up to 10,000 nodes using less GPU memory.

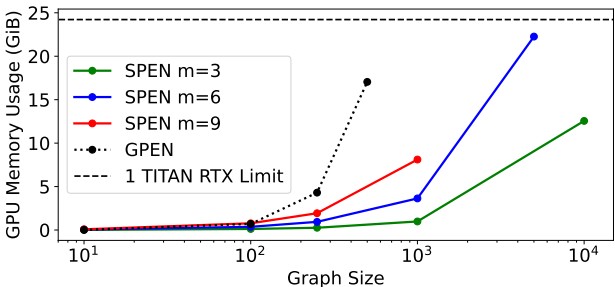

Figure 6: Computational cost of global permutation equivariant model (GPEN) and our (SPEN) model with a very similar number of model parameters for varying average size graphs. For this test we constructed random regular graphs of varying size using the NetworkX package (Hagberg et al., 2008). For SPEN sub-graphs were constructed using a 1-hop ego network policy. As is demonstrated by the log-axis, SPEN can process graphs an order of magnitude larger than global methods.

## 5 Experiments

### 5.1 Graph Benchmarks

We perform experiments with our method to compare to leading methods. For this we are looking to see how global permutation equivariant methods compare to our method? How our approach compares in terms of validation accuracy on real graph benchmarks with state-of-the-art methods? How our method scales wen compared with global permutation methods on real benchmark tasks?

**TU Datasets.** We tested our method on a series of 7 different real-world graph classification problems from the TUDatasets benchmark of (Yanardag & Vishwanathan, 2015). Five of these datasets originate from bioinformatics, while the other two come from social networks. We highlight some interesting features of each dataset. We note that both MUTAG and PTC are very small datasets, with MUTAG only having 18 graphs in the test set when using a 10% testing split. Further, the Proteins dataset has the largest graphs with an average number of nodes in each graph of 39. Also, NCI1 and NCI109 are the largest datasets having over 4000 graphs each, which one would expect to lead to less spurious results. Finally, IMDB-B and IMDB-M generally have smaller graphs, with IMDB-M only having an average number of 13 nodes in each graph. The small size of graphs coupled with having 3 classes appears to make IMBD-M a challenging problem.

**Specific comparisons to prior work.** We compare to a wide range of alternative methods, including sub-graph based methods, higher-dimensional GNN methods, and automorphism equivariant methods. We focus specifically on IGN (Maron et al., 2018) as this method uses an order-2 permutation equivariant tensor representation space for the linear map and is therefore the most similar to our base GNN model. The

Table 2: Comparison between our SPEN model and other deep learning methods. Larger mean results are better with the standard deviation around the mean given in (). Methods in comparison are: GDCNN (Zhang et al., 2018), PSCN (Niepert et al., 2016), DCNN (Atwood & Towsley, 2016), ECC (Simonovsky & Komodakis, 2017), DGK (Yanardag & Vishwanathan, 2015), DiffPool (Ying et al., 2018), CCN (Kondor et al., 2018), IGN (Maron et al., 2018), GIN (Xu et al., 2019), 1-2-3 GNN (Morris et al., 2019b), PPGN (Maron et al., 2019), LNGN (GCN) (de Haan et al., 2020), GSN (Bouritsas et al., 2020), SIN (Bodnar et al., 2021b), CIN (Bodnar et al., 2021a), and DSS-GNN (GC) (EGO) (Bevilacqua et al., 2021). All scores statistically indistinguishable (via Welch's ANOVA) from the highest mean in each benchmark have a gray background, whilst the highest mean values are in bold.

| Dataset | MUTAG | PTC | PROTEINS | NCI1 | NCI109 | IMDB-B | IMDB-M |
|---|---|---|---|---|---|---|---|
| size | 188 | 344 | 1113 | 4110 | 4127 | 1000 | 1500 |
| classes | 2 | 2 | 2 | 2 | 2 | 2 | 3 |
| avg node # | 17.9 | 25.5 | 39.1 | 29.8 | 29.6 | 19.7 | 13 |
| Results | | | | | | | |
| GDCNN | 85.8 (1.7) | 58.6 (2.5) | 75.5 (0.9) | 74.4 (0.5) | NA | 70.0 (0.9) | 47.8 (0.9) |
| PSCN | 89.0 (4.4) | 62.3 (5.7) | 75 (2.5) | 76.3 (1.7) | NA | 71 (2.3) | 45.2 (2.8) |
| DCNN | NA | NA | 61.3 (1.6) | 56.6 (1.0) | NA | 49.1 (1.4) | 33.5 (1.4) |
| DGK | 87.4 (2.7) | 60.1 (2.6) | 75.7 (0.5) | 80.3 (0.5) | 80.3 (0.3) | 67.0 (0.6) | 44.5 (0.5) |
| CCN | 91.6 (7.2) | 70.6 (7.0) | NA | 76.3 (4.1) | 75.5 (3.4) | NA | NA |
| IGN | 83.9 (13.0) | 58.5 (6.9) | 76.6 (5.5) | 74.3 (2.7) | 72.8 (1.5) | 72.0 (5.5) | 48.7 (3.4) |
| GIN | 89.4 (5.6) | 64.6 (7.0) | 76.2 (2.8) | 82.7 (1.7) | NA | 75.1 (5.1) | 52.3 (2.8) |
| PPGN v1 | 90.5 (8.7) | 66.2 (6.5) | **77.2 (4.7)** | 83.2 (1.1) | 81.8 (1.9) | 72.6 (4.9) | 50 (3.2) |
| PPGN v2 | 88.9 (7.4) | 64.7 (7.5) | 76.4 (5.0) | 81.2 (2.1) | 81.8 (1.3) | 72.2 (4.3) | 44.7 (7.9) |
| PPGN v3 | 89.4 (8.1) | 62.9 (7.0) | 76.7 (5.6) | 81.0 (1.9) | 82.2 (1.4) | 73.0 (5.8) | 50.5 (3.6) |
| LNGN (GCN) | 89.4 (1.6) | 66.8 (1.8) | 71.7 (1.0) | 82.7 (1.4) | 83.0 (1.9) | 74.8 (2.0) | 51.3 (1.5) |
| GSN-e | 90.6 (7.5) | 68.2 (7.2) | 76.6 (5.0) | 83.5 (2.3) | NA | **77.8 (3.3)** | **54.3 (3.3)** |
| GSN-v | 92.2 (7.5) | 67.4 (5.7) | 74.6 (5.0) | 83.5 (2.0) | NA | 76.8 (2.0) | 52.6 (3.6) |
| SIN | NA | NA | 76.5 (3.4) | 82.8 (2.2) | NA | 75.6 (3.2) | 52.5 (3.0) |
| CIN | 92.7 (6.1) | 68.2 (5.6) | 77.0 (4.3) | 83.6 (1.4) | **84.0 (1.6)** | 75.6 (3.7) | 52.7 (3.1) |
| DSS (EGO) | 91.5 (4.9) | 68.0 (6.1) | 76.6 (4.6) | 83.5 (1.1) | 82.5 (1.6) | 76.3 (3.6) | 53.1 (2.8) |
| SPEN | **93.3 (6.5)** | **71.3 (9.7)** | 74.8 (3.2) | **83.7 (1.5)** | 83.4 (1.2) | 75.2 (3.1) | 48.7 (2.0) |

DSS-GNN model introduced in Bevilacqua et al. (2021) is tested on multiple different sub-graph policies and here we compare to the method utilising $k$-hop ego networks as this is the most similar variant to our method.

**Implementation details and comparisons to prior work.** We utilise a 1-hop ego network sub-graph policy for all of the experiments. Further, we use a base GNN model that maps between $\rho_1 \oplus \rho_2$ permutation equivariant tensor representation space, with the final layer mapping to a $\rho_0$ permutation equivariant tensor representation space. We constrain our model to be equivariant to the automorphism groups of the bags of sub-graphs. For this we do not build a model that is equivariant to every possible automorphism group, but rather every automorphism group created by our subgraph selection policy. This was a conscious choice due to it simplifying the construction of the automorphism equivariant layer. This means that the automorphism groups extracted are each a differing size permutation group.

For MUTAG, PTC, NCI1, and NCI109 we directly constrain the model to the automorphism groups of the bags of sub-graphs. For PROTEINS, IMDB-B, and IMDB-M there exists some bags of sub-graphs which comprise of a single sub-graph. As this would lead to no weight sharing between these sub-graphs and and any other sub-graphs we parameterize the automorphism constraint to bunch bags of sub-graphs which contain few sub-graphs. Further implementation and experimental details can be found in Appendix A.6.

Table 2 compares our SPEN model to a range of other methods on benchmark graph classification tasks from TUDatasets (Morris et al., 2020). We perform statistical significance analysis using Welch's ANOVA method for comparing multiple means with different variances. We consider the null hypothesis that the means are equal and to reject this null hypothesis the p value is required to be below 0.05. We therefore make bold all the methods indistinguishable from the state-of-the-art method, see Appendix A.7 for more details on the statistical significance analysis. Comparing out method (SPEN) to a higher-dimensional global permutation equivariant (IGN) demonstrates that our method significantly outperforms the base GNN model on four

datasets and is statistically indistinguishable on the remaining three. Table 2 also highlights that our method is statistically indistinguishable from the state-of-the-art result on six out of seven datasets, and achieves a larger mean on three of these datasets. This demonstrates that our method performs competitively across the range of datasets. The strong results produced by our method suggests that our framework's improved expressivity is beneficial for learning on graph classification tasks. Further discussion on the statistical significance is provided in Appendix A.7, which highlights that previous methods which claim state-of-the-art are not achieving this in a statistically meaningful way.

Figure 7 demonstrates that the improved scalability of our methods on regular graphs carries over onto graphs on real-world benchmarks. This demonstrates that our method offers a significant improvement in scalability over global permutation equivariant models.

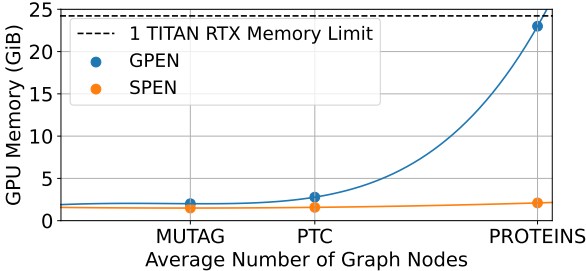

Figure 7: Computational cost of a global permutation equivariant model (GPEN) and our method (SPEN) with a very similar number of model parameters and batch size for datasets with varying average size graphs from the TUDatasets. For SPEN sub-graphs were constructed using a 1-hop ego network policy.

# 6 Future Work

From Table 2 it is clear that IMDB-M is a dataset for which our method has weaker performance. As stated in Section A.6.2 between hidden layers in our network, for the experiments in this paper, we only make use of order 1 and 2 representations. As it was shown by Maron et al. (2019) that increasing the order of the permutation representation increases the expressivity in line with the $k$-WL test, the expressivity of our method could be improved through the consideration of higher order permutation representations. Further, we parameterized the automorphism constraint in IMDB-M and therefore exploring alternative parameterizations of this constraint could lead to improved results.

# 7 Conclusion

We present a graph neural network framework for building models that operate on $k$-ego network sub-graphs that respects both the permutation symmetries of individual sub-graphs and is equivariant to the automorphism groups across bags of sub-graphs. The choice of sub-graph policy leads to a novel choice of automorphism groups for the bags of sub-graphs. The framework is more scalable than global higher-dimensional GNNs through the use of sub-graphs and we have both theoretically and experimentally demonstrated this. We have shown that SPEN is provably more expressive than the base higher-dimensional permutation equivariant GNN and sub-graph MPNNs through the choice sub-graph selection policy, permutation equivariant base GNN, and automorphism equivariant kernel constraint. We have provided theoretical analysis demonstrating the expressivity of the framework. Finally, we have shown that SPEN performs competitively across multiple graph classification benchmarks, achieving statistically indistinguishable accuracy compared to the state-of-the-art method on six out of seven datasets. We believe that our framework is a step forward in the development of graph neural networks, demonstrating theoretically provable expressivity, scalability, and experimentally achieving strong performances on benchmark datasets.

**Funding and Acknowledgements**

Joshua Mitton is supported by a University of Glasgow Lord Kelvin Adam Smith Studentship. Roderick Murray-Smith is grateful for EPSRC support through grants EP/R018634/1 and EP/T00097X/1.

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

## A Appendix

### A.1 Mathematical Background

#### A.1.1 Group Theory

**Definition A.1.** A group is a set $G$ with a binary operation $\circ$, usually denoted $(G, \circ)$ satisfying the following laws:
(G0) (Closure law): For all $g, h \in G$, $g \circ h \in G$.
(G1) (Associative law): $g \circ (h \circ k) = (g \circ h) \circ k$ for all $g, h, k \in G$.
(G2) (Identity law): There exists $e \in G$ such that $g \circ e = e \circ g = g$ for all $g \in G$.
(G3) (Inverse law): For all $g \in G$, there exists $h \in G$ with $h \circ g = g \circ h = e$.

Definition A.1 provides a definition of a group, where a group is commonly written as $(G, \circ)$, although where the binary operation is not ambiguous we will write a group as $G$. The commutative law is not included within the definition of a group, although if a group also satisfies $g \circ h = h \circ g$ for all $g, h \in G$ then the group is called commutative or abelian.

**Example A.2.** Real numbers and addition.
The set $\mathbb{R}$ together with the binary operation $+$ yields a group $(\mathbb{R}, +)$. This is the group of real numbers with the binary operation of addition.

To show that $(\mathbb{R}, +)$ in Example A.2 is a group we need to consider each of the axioms.
(G0) For all $g, h \in \mathbb{R}$ we need to check that $g + h \in \mathbb{R}$. From the definition of $\mathbb{R}$ is clear that addition of two number yields a real number.
(G1) The order of summation of real numbers does not impact the result and hence $g + (h + k) = (g + h) + k$ for all $g, h, k \in \mathbb{R}$, so it is associative.
(G2) It is known that adding zero to a real number does not change it, hence $0 \in \mathbb{R}$ satisfies $g + 0 = 0 + g = g$ for all $g \in \mathbb{R}$, and it has an identity.
(G3) Finally, for each real number there exists the negative equivalent, such that for all $g \in \mathbb{R}$ there exists $-g \in \mathbb{R}$ satisfying $-g + g = g + -g = 0$.

Therefore we can conclude that $(\mathbb{R}, +)$ is a group.

**Definition A.3.** Subgroup
Given a group $G$, a sub-group of $G$ is a subset of $G$ which, using the same operation as in $G$, is itself a group. A subgroup $H$ of $G$ is denoted by $H \leq G$.

**Definition A.4.** Group homomorphism.
Given two groups $G$ and $H$, a homomorphism $\theta : G \to H$ is a function $\theta$ from $G$ to $H$ that satisfies the condition

$$(g_1 g_2)\theta = (g_1 \theta)(g_2 \theta) \ \forall g_1, g_2 \in G.$$

A homomorphism that is one-to-one and onto is called an isomorphism. $G$ and $H$ are called isomorphic is there is an isomorphism between the two groups. Two groups being isomorphic means that from the point of view of abstract algebra they are the same, even if their elements are completely different.

#### A.1.2 Category Theory

This section does not provide a complete overview of category theory, nor even a full introduction, but aims to provide a sufficient level of understanding to aid the reader with further sections of the paper, where we believe presenting the comparison between models from a category theory perspective makes more clear the distinctions between them. A category, $\mathcal{C}$, consists of a set of objects, $\mathrm{Ob}(\mathcal{C})$, and a set of morphisms (structure-preserving mappings) or arrows, $f : A \to B$, $A, B \in \mathrm{Ob}(\mathcal{C})$. There is a binary operation on morphisms called composition. Each object has an identity morphism. Categories can be constructed from given ones by constructing a subcategory, in which each object, morphism, and identity is from the original category, or by building upon a category, where objects, morphisms, and identities are inherited from the original category. A functor is a mapping from one category to another that preserves the categorical structure.

For two categories $\mathcal{C}$ and $\mathcal{D}$ a functor $F : \mathcal{C} \to \mathcal{D}$ maps each object $A \in \mathrm{Ob}(\mathcal{C})$ to an object $F(A) \in \mathrm{Ob}(\mathcal{D})$ and maps each morphism $f : A \to B$ in $\mathcal{C}$ to a morphism $F(f) : F(A) \to F(B)$ in $\mathcal{D}$.

**Definition A.5.** A *groupoid* is a category in which each morphism is invertible. A groupoid where there is only one object is usually a group.

## A.2   WL Variants and Proofs for Section 4

This section outlines the necessary definitions and proofs to support the theoretical analysis in Section 4. The expressivity analysis follows from prior work of (Bevilacqua et al., 2021) and (Zhao et al., 2021).

**Definition A.6.** (Vertex coloring). A vertex coloring is a function mapping a graph and one of its nodes to a "color" from a fixed color palette (Rattan & Seppelt, 2021; Bevilacqua et al., 2021).

Generally, a vertex coloring is a function $c : \mathcal{V} \to C, (G, v) \mapsto c_v^G$, where $\mathcal{V}$ is the set of all possible tuples of the form $(G, v)$ with $G = (V, E)$ the set of all finite graphs and $v \in V$.

**Definition A.7.** (Vertex color refinement). Let $c, d$ be two vertex colorings. We say that $d$ refines $c$ when for all graphs $G = (V_G, E^G)$, $H = (V^H, E^H)$ and all vertices $v \in V^G$, $u \in V^H$ we have that $g_v^G = d_u^H \Rightarrow c_v^G = c_u^H$. We write $d \sqsubseteq c$.

When working with a specific graph pair $G^1$, $G^2$, the refinement $d$ of $c$ is written $d \sqsubseteq_{G^1, G^2} c$, when, in particular, it holds that $\forall v \in V^{G^1}, u \in V^{G^2}, d_v^{G_1} = d_u^{G_2} \Rightarrow c_v^{G_1} = c_u^{G_2}$.

The 1-WL test represents a graph as a multiset (or histogram) of colors associated with its nodes. This coloring induces a partitioning of the nodes into color classes, where two nodes belong to the same partition is and only if they have the same coloring. The algorithm starts from some initial coloring and iteratively updates the coloring, leading to at each step, where the algorithm does not terminate, a finer-grained node partitioning. Each of these iterations is a refinement step, since, if $c$ indicates the coloring computed at iteration $t$ then the subsequent coloring at iteration $t + 1$ is given by $c^{t+1} \sqsubseteq_{G, G} c^t$

**Definition A.8.** (Sub-graph-1-WL (Zhao et al., 2021)). Sub-graph-1-WL generalises the 1-WL test by replacing the color refinement step $c_v^{t+1} = \mathrm{HASH}(Star^t(v))$ with $c_v^{t+1} = \mathrm{HASH}(\mathrm{G}^{\mathrm{t}}[\mathcal{N}_{\mathrm{k}}(\mathrm{v})]), \forall \mathrm{v} \in \mathcal{V}$. Where $G[\mathcal{N}_k(v)]$ is the $k$-hop egonet.

We start by proving that sub-graph-WL is at least as expressive as 1-WL. For this we first characterise our sub-graph-WL to make the comparison between a refinement strategy for a bag of sub-graphs and those which operate on graphs.

**Definition A.9.** (Sub-graph-WL node refinement). For a graph $G = (V, E)$ we denote $S_G$ as a bag of sub-graphs generated by taking the $k$-hop ego net of each node $v \in V$. The color refinement for node $v$ at time step $t \geq 0$, $C_v^t$, is given by the set of node colors across the sub-graphs, denoted as $\{\{c_{v,H}^t\}\}_{H \in S_G}$.

**Lemma A.10.** $b \sqsubseteq a$. *That for all graphs $G^1 = (V^1, E^1)$ and $G^2 = (V^2, E^2)$ and all nodes $v \in V^1$, $w \in V^2$ that $b_v = b_w \Rightarrow a_v = a_w$.*

*Proof.* For such a node refinement policy, inclusive of node refinement across sub-graphs, Bevilacqua et al. (2021) show that, for $b$ the node coloring from a sub-graph-WL refinement and $a$ the node coloring from a WL refinement, $b \sqsubseteq a$. It then follows that for all graphs $G^1 = (V^1, E^1)$ and $G^2 = (V^2, E^2)$ and all nodes $v \in V^1$, $w \in V^2$ that $b_v = b_w \Rightarrow a_v = a_w$.

**Lemma A.11.** *Sub-graph-WL is at least as powerful as sub-graph-1-WL in distinguishing non-isomorphic graphs.*

*Proof.* We denote $a$ colorings by the sub-graph-1-WL algorithm, $b$ colorings on each sub-graph by the sub-graph-WL algorithm, and $c$ coloring on each node within a sub-graph by the sub-graph-WL algorithm. We also denote $S^1$, $S^2$ as the bags of sub-graphs from $G^1$, $G^2$ respectively. If $|S^1| \neq |S^2|$ then the two graphs are trivially distinguished by sub-graph-WL. In the case where $|S^1| = |S^2|$ we seek to show that if sub-graph-1-WL (Zhao et al., 2021) identifies non-isomorphic graphs then so does sub-graph-WL.

First recall that sub-graph-1-WL at time step $t$ deems two graphs non-isomorphic if the following two are assigned two different multisets of node colors:

$$\{\{a_v^t | v \in \mathcal{V}^1\}\} \neq \{\{a_w^t | w \in \mathcal{V}^w\}\},$$

while sub-graph-WL deems them non-isomorphic when the following two are assigned two different multisets of subgraph colors:

$$\{\{b_{S_k^1}^t\}\}_{k=1}^m \neq \{\{b_{S_h^2}^t\}\}_{h=1}^m.$$

If it is given that sub-graph-1-WL distinguishes between two graphs at iteration $T$, then by Lemma A.10 $b^T \sqsubseteq a^T$. In addition, Bevilacqua et al. (2021) prove that for such a coloring at $T$ a sub-graph refinement policy such as sub-graph-WL is refined by the coloring generated at $T+1$ on any pair of sub-graphs: $\forall H_1, H_2 \in S^1 \cup S^2 \ c^{T+1} \sqsubseteq_{H_1, H_2} b^T$. The proof follows from the definition of the refinement step in an algorithm for a bag of sub-graphs, namely, the inclusion of a term which refines over the multiset of node colors across sub-graphs implies that if $C_v^T = C_u^T$ then $b_v^T = b_u^T$. This gives that if sub-graph-1-WL can distinguish between two graphs at time step $T$ then the sub-graph refinement policy yields distinct colors to any pair of sub-graphs. Therefore, sub-graph-WL can distinguish between two graphs that sub-graph-1-WL can and is at least as expressive.

This provides the necessary detail for the proof of Theorem 4.2. To prove that sub-graph-WL is strictly more powerful than 1&2-WL we could instead prove that sub-graph-1-WL is strictly more powerful than 1&2-WL and then by Lemma A.11 the proof that sub-graph-WL is strictly more powerful than 1&2-WL is complete. In fact, Zhao et al. (2021) prove that sub-graph-1-WL is strictly more powerful than 1&2-WL by presenting a pair of non-ismorphic graphs that sub-graph-1-WL distinguishes but 1-WL cannot. Therefore, we can conclude that our sub-graph-WL is strictly more powerful than 1&2-WL.

## A.3 Previous Methods

### A.3.1 Global Equivariant Graph Networks

**Global Permutation Equivariance**. Global permutation equivariant models have been considered by Hartford et al. (2018); Maron et al. (2018; 2019); Albooyeh et al. (2019), with Maron et al. (2018) demonstrating that for order-2 layers there are 15 operations that span the full basis for an permutation equivariant linear layer. These 15 basis elements are shown in Figure 5 with each basis element given by a different color in the map from representation $\rho_2 \rightarrow \rho_2$. Despite these methods, when solved for the entire basis space, having expressivity as good as the $k$-WL test, they operate on the entire graph. Operating on the entire graph features limits the scalability of the methods. In addition to poor scalability, global permutation appears to be a strong constraint to place upon the model. In the instance where the graph is flattened and an MLP is used to update node and edge features the model would have $n^4$ trainable parameters, where $n$ is the number of nodes. On the other hand, a permutation equivariant update has only 15 trainable parameters and in general $15 \ll n^4$.

Viewing a global permutation equivariant graph network from a category theory perspective there is one object with a collection of arrows representing the elements of the group. Here the arrows or morphisms go both from and to this same single object. The feature space is a functor which maps from a group representation to a vector space. For a global permutation equivariant model the same map is used for every graph.

Symmetric Group

**Global Naturality** Global natural graph networks (GNGN) consider the condition of naturality, (de Haan et al., 2020). GNGNs require that for each isomorphism class of graphs there is a map that is equivariant

to automorphisms. This naturality constraint is given by the condition $\rho'(\phi) \circ K_G = K_{G'} \circ \rho(\phi)$, which must hold for every graph isomorphism $\phi : G \to G'$ and linear map $K_G$. While the global permutation equivariance constraint requires that all graphs be processed with the same map, global naturality allows for different, non-isomorphic, graphs to be processed by different maps and as such is a generalisation of global permutation equivariance. As is the case for global permutation equivariant models, GNGNs scale poorly as the constraint is placed over the entire graph and linear layers require global computations on the graphs.

Viewing a GNGN from a category theory perspective there is a different object for each concrete graph, which form a groupoid. Then, there is a mosphism or arrow for each graph isomorphism. These can either be automorphisms, if the arrow maps to itself, or isomorphisms if the arrow maps to a different object. The feature spaces are functors which map from this graph category to the category of vector spaces. The GNG layer is a natural transformation between such functors consisting of a different map for each non-isomorphic graph.

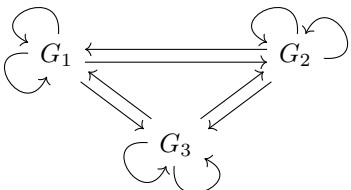

Groupoid of Concrete Graphs

### A.3.2 Local Equivariant Graph Networks

Local equivariant models have started to receive attention following the successes of global equivariant models and local invariant models. The class of models that are based on the WL test are not in general locally permutation equivariant in that they still use a message passing model with permutation invariant update function. Despite this, many of these models inject permutation equivariant information into the feature space, which improves the expressivity of the models (Bouritsas et al., 2020; Morris et al., 2019a; Bodnar et al., 2021b;a). The information to be injected into the feature space is predetermined in these models by a choice of what structural or topological information to use, whereas our model uses representations of the permutation group, making it a very general model that still guarantees expressivity.

In contrast to utilising results from the WL test covariant compositional networks (CCN) look at permutation equivariant functions, but they do not consider the entire basis space as was considered in Maron et al. (2018) and instead consider four equivariant operations (Kondor et al., 2018). This means that the permutation equivariant linear layers are not as expressive as those used in the global permutation equivariant layers. Furthermore, in a CCN the node neighbourhood and feature dimensions grow with each layer, which can be problematic for larger graphs and limits their scalability. Another local equivariant model is that of local natural graph networks (LNGN) (de Haan et al., 2020). An LNGN uses a message passing framework, but instead of using a permutation invariant aggregation function, it specifies the constraint that node features transform under isomophisms of the node neighbourhood and that a different message passing kernel is used on non-isomorphic edges. In practice this leads to little weight sharing in graphs that are quite heterogeneous and as such the layer is re-interpreted such that a message from node $p$ to node $q$, $k_{pq}v_p$, is given by a function $k(G_{pq}, v_p)$ of the edge neighbourhood $G_{pq}$ and feature value $v_p$ at $p$. In comparison to our method LNGN amounts to choosing a different automorphism group, where a LNGN is equivariant to the edge sub-graph when performing the message passing. On the other hand, we utilise the $k$-ego network sub-graphs of each node in the graph. As was noted by the authors of LNGN their choise of automorphism group leads to little weight sharing and requires parameterising. While it is still possible for some datasets, namely those which are particularly heterogeneous, that our choice of automorphism group requires parameterising, in practise this was not required on most datasets, which allows us to use the true automorphism group. We also utilise a different base update function in comparison to LNGNs, where we use a higher-order permutation equivariant update function and a LNGN uses a GCN.

Viewing a LNGN from a category theoretic perspective there is a groupoid of node neighbourhoods where morphisms are isomorphisms between node neighbourhoods and a groupoid of edge neighbourhoods where morphisms are ismorphisms between edge neighbourhoods. In addition, there is a functor mapping from edge neighbourhoods to the node neighbourhood of the start node and a functor mapping similarly but to the tail node of the edge neighbourhood. The node feature spaces are functors mapping from the category of node neighbourhoods to the category of vector spaces. Further, composition of two functors creates a mapping from edge neighbourhoods to the category of vector spaces. A LNG kernel is a natural transformation between these functors.

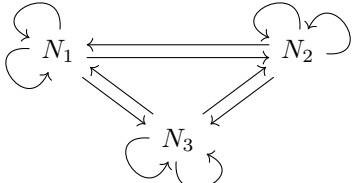

Groupoid of Node Neighbourhoods

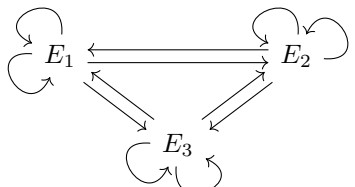

Groupoid of Edge Neighbourhoods

Another local equivariant graph network, which makes use of an automorphism group symmetry, is that of Autobahn (Thiede et al., 2021), where this approach uses the automorphism groups of cycles and paths. This choice again differs from ours. A key difference can be seen that our method is general in the way it can be used on any graph dataset while Autobahn is designed specifically for molecular datasets. Our choice of automorphism group and sub-graph selection policy ensures that the information of every node is made use of in the model, which Autobahns choice of automorphism group can lead to nodes being ignored.

Another method, which does not make use of automorphism group equivariance, makes use of sub-groups and graph symmetries, is ESAN (Bevilacqua et al., 2021). This work explores multiple sub-graph selection policies and is therefore different from our work in that we chose one specific sub-graph selection policy. Our choice of sub-graph selection policy leads to a natural choice of automorphism group, which requires less parameterisation than previous works and improves scalability. On the other hand, ESAN does not consider automorphism groups and therefore this is not a consideration of their work and their method focuses on comparing the expressivity of different sub-graph selection policies.

In addition, $k$-reconstruction GNNs (Cotta et al., 2021) also consider sub-graphs, although here vertex removed sub-graphs are considered, which is different to the sub-graph choice we made. Removing single nodes from a graph would not yield the same improvement in scalability as we demonstrate in our method due to each sub-graph being almost the same size as the original graph, while we demonstrated using $k$-ego network sub-graphs yields sub-graphs which are much smaller than the original graph in practise. Furthermore, in our work we show how the combination of sub-graph selection policy, automorphism equivariance constraint and higher-order permutation GNN update functions improves expressivity beyond 1-WL, while $k$-reconstruction GNNs compare to 1-WL.

Finally, the GRAPE model (Xu et al., 2021) is somewhat similar to Autobahn in that they use sub-graph templates to select the automorphism constraint, although it appears that for GRAPE this is chosen to be more general than for Autobahn. This still differs from our sub-graph selection policy and has to be pre-determined before building a model. On the other hand, our method, using a $k$-ego network policy, has

the automorphism constraint driven by the data, ensuring the approach is applicable across a range of graph datasets.

### A.4 Implementing other models within our framework

In the datasets used, for graph classification benchmark tasks, the input to the model is a graph with node and edge features, this can be represented as $2^{\text{nd}}$ order permutation representation, so the input representation would be $j = 2$. The convolution can then map from this representation, $\rho_j$, to multiple different representation spaces, $\rho_0 \oplus \rho_1 \oplus \cdots \oplus \rho_i$. Subsequent convolutions can then map from these multiple permutation representations, $\rho_0 \oplus \rho_1 \oplus \cdots \oplus \rho_i$, to multiple different permutation representations, $\rho_0 \oplus \rho_1 \oplus \cdots \oplus \rho_i$. The choice of representations used can be made depending on a trade off between expressivity and computational cost, as lower order representation spaces have less expressivity, but also lower computational cost.

**Local Natural Graph Networks** (LNGNs) (de Haan et al., 2020) take the input feature space and embed this into an invariant scalar feature of the edge neighbourhood graph. This is the same as using specific choice $k$-hop sub-graph creation and permutation representation space for the sub-graph convolution. In the case of LNGNs the choice would be $k = 1$ and mapping the input feature space to representation $\rho_0$ creating a permutation invariant feature space. Then any graph neural network with invariant features can be used, in the paper the choice made is to use a GCN (Kipf & Welling, 2016), which can also be covered by our framework. Here the choice would again be to use $k = 1$ when creating the subgroups and using a subgraph convolution with representation spaces $\rho_0 \rightarrow \rho_0$.

**Global Equivariant Graph Networks** (EGNs) (Maron et al., 2018) use a choice of $k = n$, for $n$-node graphs when creating the sub graphs, which corresponds to not selecting a sub graph and instead operating over the entire graph. They then use the representation space $\rho_2 \rightarrow \rho_2$ mapping from a graph feature space to a graph feature space.

**Local Permutation Equivariant Graph Networks** (LPEGN) (Ours) In our paper we choose to use $k = 1$ throughout to keep inline with the vast majority of previous work on graph neural networks, but we use a representation space of $\rho_1 \oplus \rho_2 \rightarrow \rho_1 \oplus \rho_2$ in the hidden layers of the model and we note that this was simply a choice that seemed a simple case to present as a comparison with previous work in the benchmark classification task.

### A.5 Architectural Details of the SPEN Framework

The main figure outlining the model concept is provided in Figure 2. The first stage in the model is to split the input graph into a bag of sub-graphs. To do this we utilise a $k$-ego network policy, where for each node in the input graph we extract the neighbouring nodes and the induced connectivity of these nodes up to $k$-hop away from the initial node. This induced connectivity sub-graph is then extracted and becomes one of the sub-graphs within the bag. This process is repeated for each node in the input graph, creating a bag of sub-graphs. This can be seen in step (2) in Figure 2.

At this point the input graph is represented as a bag of sub-graphs. The next step in the model is to split these into there corresponding automorphism groups. As these are sub-graphs with permutation symmetry the automorphism groups are defined as

$$\text{Aut}(H) = \{\sigma \in \mathbb{S}_n | A^\sigma = A\}, \tag{2}$$

where $A$ is the adjacency matrix of $H$ and $\sigma$ is a permutation action on the sub-graph. Due to the choice of sub-graph selection policy, where the induced connectivity is extracted, we can consider missing edges in the sub-graph as zero feature edges. This is the same approach as would be taken in Maron et al. (2018) and all approaches which operated on a dense adjacency matrix of the graph. Therefore, each sub-graph automorphism group is fully determined by the size of the sub-graph, namely the number of nodes in the sub-graph. Then each sub-graph is placed in a bag of sub-graphs such that each sub-graph within the bag belongs to the same automorphism group. These two steps of finding the sub-graphs and placing into bags of

sub-graphs such that each bag corresponds to a single automorphism group can be combined into one step. This is achieved by placing the sub-graphs into the correct automorphism group bag of sub-graphs as each sub-graph is extracted. This can be seen as moving to step (3) in Figure 2.

Now the input graph is represented by multiple bags of sub-graphs, each bag corresponding to a different automorphism group. Next we considered step (3) in Figure 2. As our model operated on sub-graphs we use a graph neural network architecture. Here we choose to use permutation equivariant neural networks and utilise a general approach for operating on higher-order objects. A general recipe for building group equivariant neural networks was provided in Kondor & Trivedi (2018). Following this formalism, we treat any object that transforms under a group action as a function on the group. In the case of an object which transforms under a 0-order permutation this would correspond to a single node, and an example of this would be a graph after being pooled. Here, in the case of a single feature dimension, there is just a single weight and this is an uninteresting case where there is actually nothing to permute. An object which transforms under a 1-order permutation is a set and an object which transforms under a 2-order permutation is a graph. This concept can be extended to objects which transform under higher order permutations. In addition, we can map between different order permutations. Enforcing permutation equivariance within a neural network layer mapping between an input and output object which we require to transform under a permutation group action places a restriction over the weights in the model. It is possible to find bases for a mapping between different objects transforming under different order permutation transformations, which provides the number of permissible weights for a single feature dimension neural network. We provide an example of some of these bases functions in Figure 5. Here we use the notation $\rho_i$ to represent an object which transforms under an $i$-order permutation transformation. Following this, we use the notation $\rho_i \to \rho_j$ to denote a mapping between an object transforming under an $i$-order permutation transformation and an object transforming under an $j$-order permutation transformation. We have used the notation $\rho$ due to this being the common notation to use for representations in group theory. Kondor & Trivedi (2018) defined the group convolution and made the connection to Fourier analysis, where the function is decomposed into irreducible representations. These irreducible representations can be combined using the direct sum to create other group representations, for example $\rho_i = \rho_a \oplus \rho_b$. Here we are making use of permutation representations to restrict the space of the linear update function such that we use the bases shown in Figure 5, and hence the use of $\rho$.

Now that we have the general recipe for constructing higher order permutation equivariant graph neural networks, we consider the specifics of the linear update function used within our model. In our model we construct a graph neural network which comprises of mappings between objects which transform under different order permutation transformations. Our model uses a different set of weights to perform the linear update mapping for each automorphism group. This can be viewed as building a separate graph neural network for each automorphism group; despite this, the choice of mappings, i.e. the feature dimension and order of objects, is kept the same for each automorphism group. This different set of weights which performs the linear mapping within the graph network is symbolised in Figure 5 by showing three GNNs for automorphism groups A2, A3, A4, for this example graph. This explains the linear map (4) which produces the outputs (5) in the figure.

At this point, the key concepts of the model are explained, namely the splitting of the graph into sub-graphs which are stored in separate bags for each automorphism group, the core GNN update functions which comprise of higher-order permutation update functions, and the automorphism constraint placed over the model via the enforced weight sharing in the model. Following this, there is an averaging of node and edge features across the sub-graphs. This comprises the linear update function of the model and a choice of non-linearity can be used. In our experiments we used the ELU non-linearity. The entire model is composed by stacking multiple of these layers and, in the case of a graph classification task, adding a pooling layer. In the notation of our framework a set or graph pooling layer is a map from an object which transforms under a 1 or 2-order permutation transformation to a 0-order permutation transformation respectively.

Table 3: Different range of graph sizes and sub-graph sizes for each dataset considered from TUDatasets.

| Dataset | MUTAG | PTC | PROTEINS | NCI1 | NCI109 | IMDB-B | IMDB-M |
|---|---|---|---|---|---|---|---|
| Graph Sizes | 10-28 | 2-109 | 4-620 | 3-111 | 4-111 | 12-136 | 7-89 |
| Sub-graph Sizes | 2-5 | 2-5 | 1-26 | 2-5 | 1-6 | 1-135 | 1-88 |
| Mean Sub-graph Size | 3.2 | 3.0 | 4.7 | 3.2 | 3.2 | 9.8 | 10.1 |

### A.6  Implementation Details and Datasets

### A.6.1  TUDatasets

We present the range of graph sizes and sub-graph sizes when utilising a 1-ego network sub-graph extraction policy in Table 3.

### A.6.2  Model Architecture

We consider the input graphs as an input feature space that is an order 2 representation. For each local permutation equivariant linear layer we use order 1 and 2 representations as the feature spaces. This allows for projection down from graph to node feature spaces through the basis for $\rho_2 \to \rho_1$, projection up from node to graph feature spaces through the basis for $\rho_1 \to \rho_2$, and mappings across the same order representations through $\rho_2 \to \rho_2$ and $\rho_1 \to \rho_1$. The final local permutation equivariant linear layer maps to order 0 representations through $\rho_2 \to \rho_0$ and $\rho_1 \to \rho_0$ for the task of graph level classification. In addition to the graph layers, we also add 3 MLP layers to the end of the model.

Despite these specific choices which were made to provide a baseline of our method for comparison to existing methods the framework we present is much more general and different representation spaces can be chosen. Therefore, different permutation representation spaces, $\rho_1 \oplus \rho2 \oplus \cdots \oplus \rho_i$, can be chosen for different layers in the model and a different $k$ value can be chosen when creating the sub-graphs.

We also present an algorithm for the model below:

Input graph $G$
**for** $i \leftarrow 1$ to $K$ **do**
    Extract $k$ ego network subgraph $H_i$
    Place $H_i$ in bag $S_{H^{|H_i|}}$
**end for**
**for each** $S_H$ **do**
    **for** $H_i$ in $S_H$ **do**
        $H_i' \leftarrow f^{S_H}_{0(\rho_2 \to \rho_2)}(H_i)$
        $N_i' \leftarrow f^{S_H}_{0(\rho_2 \to \rho_1)}(H_i)$
    **end for**
**end for**
**for** layers $l \leftarrow 1$ to $L-1$ **do**
    Pool features across subgraphs $G' \leftarrow S_H$
    **for** $i \leftarrow 1$ to $K$ **do**
        Extract $k$ ego network subgraph $H_i$
        Place $H_i$ and $N_i$ in bag $S_{H^{|H_i|}}$
    **end for**
    **for each** $S_H$ **do**
        **for** $H_i$ in $S_H$ **do**
            $H_i' \leftarrow f^{S_H}_{l(\rho_2 \to \rho_2)}(H_i)$
            $N_i' \leftarrow f^{S_H}_{l(\rho_2 \to \rho_1)}(H_i)$
            $H_i' \leftarrow f^{S_H}_{l(\rho_1 \to \rho_2)}(N_i)$

$$N_i' \leftarrow f_{l(\rho_1 \to \rho_1)}^{S_H}(N_i)$$
**end for**
**end for**
**end for**
**for each** $S_H$ **do**
    **for** $H_i$ in $S_H$ **do**
$$G_i' \leftarrow f_{L(\rho_2 \to \rho_0)}^{S_H}(H_i)$$
$$G_i' \leftarrow f_{L(\rho_1 \to \rho_0)}^{S_H}(H_i)$$
    **end for**
**end for**
`Pool graph features across subgraph bags` $G' \leftarrow S_H$
`Update and predict graph classification target with an MLP model`

Where each function $f$ is a function update with bases set given in Figure 5.

### A.6.3 Implementation details

For all experiments we used a 1-hop ego networks as this provides the most scalable version of our method. We trained the model for 50 epochs on all datasets using the Adam optimizer. We considered the evaluation procedure as was conducted in (Bevilacqua et al., 2021; Xu et al., 2019; Yanardag & Vishwanathan, 2015; Niepert et al., 2016). Specifically, we conducted 10-fold cross validation and reported the average and standard deviation of validation accuracies across the 10 folds. For all datasets we use 6 automorphism equivariant layers with base GNN utilising $\rho_1 \oplus \rho_2$ representation space.

### A.6.4 Sub-graph Compute Run-time

The current implementation of the model computes the sub-graphs on the fly, although this could be moved into a pre-processing stage which would speed up run-time of the model and slow down the pre-processing stage. Here, in Table 4, we provide the run-time of the computation of the sub-graphs for each dataset to provide an idea of how long this process takes in our 1-hop SPEN model.

Table 4: Run-time to compute sub-graphs for each dataset from TUDatasets when using a 1-hop ego-net sub-graph selection policy.

| Dataset | Sub-graph Compute Run-time [s] |
| --- | --- |
| MUTAG | 0.90 |
| PTC | 2.26 |
| PROTEINS | 14.25 |
| NCI1 | 31.21 |
| NCI109 | 30.96 |
| IMDB-B | 17.79 |
| IMDB-M | 19.75 |

### A.7 Further Results Discussion

In addition to the comparison across datasets in Table 2 Figures 8, 9, 10, 11, 12, 13, and 14 show the test accuracy distribution of the SPEN method and compares to other methods from Table 2. This shows for the smaller datasets that the spread of test accuracy's is larger leading to our method and others presenting large standard deviations over the results. Comparing the SPEN results to the other methods here highlights that the SPEN result is competitive across a range of datasets. For the NCI1 and NCI109 datasets the distribution of results of our method highlights the strong performance of the SPEN method. For IMDBB and IMDBM the distribution of results for the SPEN method also highlights that it is competitive on these datasets.

We also present analysis of the statistical significance of test accuracies from each method across the seven datasets considered. Here we make use of the Welch's ANOVA method for comparing the means of multiple scores with different variances. We consider the null hypothesis that the means are equal and to reject this null hypothesis the p value is required to be below 0.05. Tables 5 and 6 show that for leading methods SPEN, CIN, GSN, CCN, and DSS the p values between each of these methods is greater than 0.05 and therefore we cannot reject the null hypothesis, and thus conclude that the means are not significantly different. Despite this our SPEN method does produce statistically significantly better results than some benchmark results. Table 7 compares the statistical significance of results on the Proteins dataset, which is one where our method appears to perform more poorly ranking 15th for mean values. Here we show that when comparing the leading results of PPGN, CIN, IGN, DSS, GSN, and SIN there is no statistical significance between our mean accuracy and theirs. Therefore we can conclude that there is no significant difference between the means and our method is comparable with SOTA results. Tables 8, 9, and 10 also show that for leading methods the p values between each of these methods is greater than 0.05 and therefore we cannot reject the null hypothesis, and thus conclude that the means are not significantly different. Despite this our SPEN method does produce statistically significantly better results than some benchmark results. Finally, Table 11 does show statistically significant results between our method and leading methods and therefore our method is under-performing on this dataset. This is something we seeks to explore and improve in the future. Overall the analysis of the statistical significance of results by our SPEN method and other benchmark results highlights that the mean accuracy's from recent leading methods are not significantly different. This is the same across each recent method claiming SOTA on some benchmark datasets and not an exclusive result to our method. This highlights that our method is competitive with SOTA methods across a range of benchmarks.

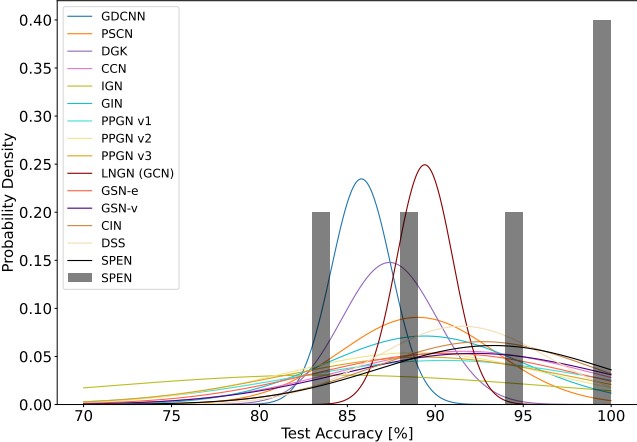

Figure 8: Comparison between our SPEN method and other methods on the MUTAG dataset. Results for the SPEN method are also presented as a histogram of the 10-fold runs. Each other method is given as a Gaussian distribution with mean and standard deviation as is presented in Table 2.

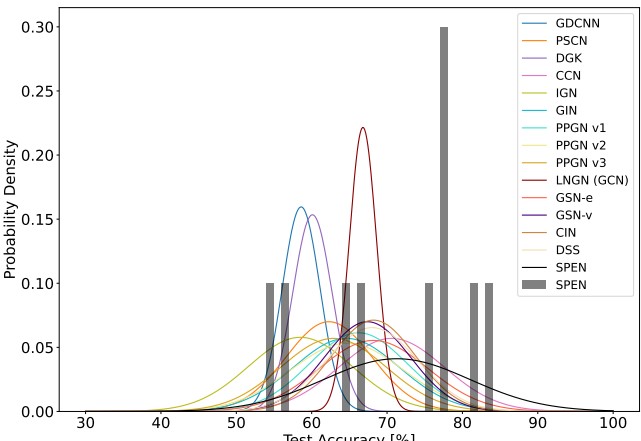

Figure 9: Comparison between our SPEN method and other methods on the PTC dataset. Results for the SPEN method are also presented as a histogram of the 10-fold runs. Each other method is given as a Gaussian distribution with mean and standard deviation as is presented in Table 2.

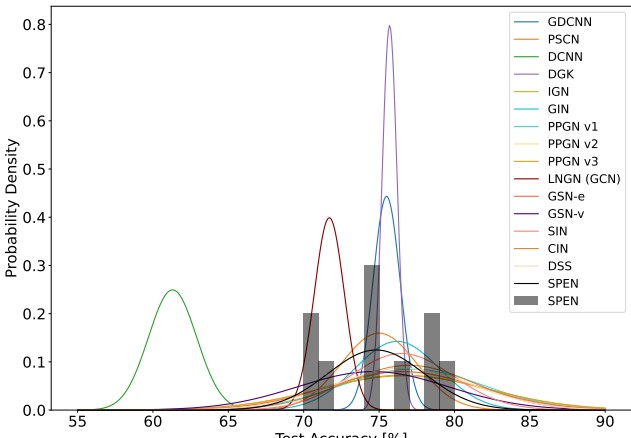

Figure 10: Comparison between our SPEN method and other methods on the PROTEINS dataset. Results for the SPEN method are also presented as a histogram of the 10-fold runs. Each other method is given as a Gaussian distribution with mean and standard deviation as is presented in Table 2.

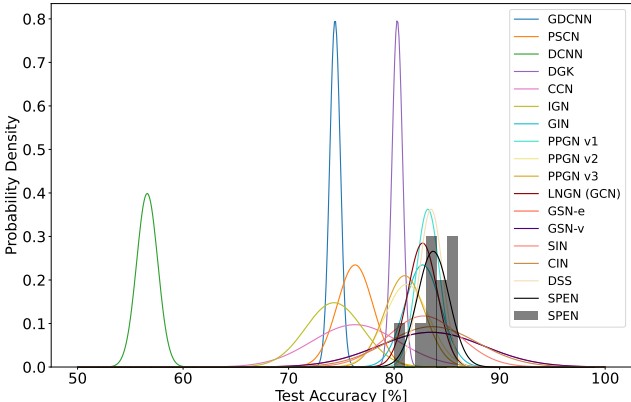

Figure 11: Comparison between our SPEN method and other methods on the NCI1 dataset. Results for the SPEN method are also presented as a histogram of the 10-fold runs. Each other method is given as a Gaussian distribution with mean and standard deviation as is presented in Table 2.

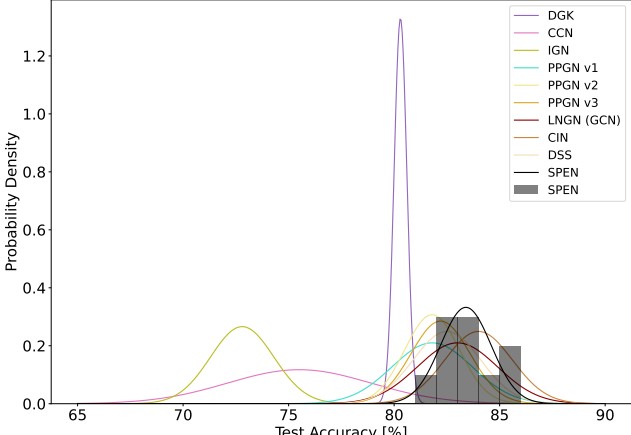

Figure 12: Comparison between our SPEN method and other methods on the NCI109 dataset. Results for the SPEN method are also presented as a histogram of the 10-fold runs. Each other method is given as a Gaussian distribution with mean and standard deviation as is presented in Table 2.

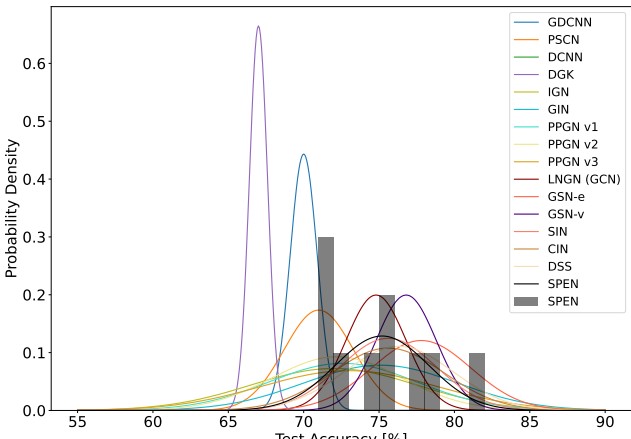

Figure 13: Comparison between our SPEN method and other methods on the IMDB-B dataset. Results for the SPEN method are also presented as a histogram of the 10-fold runs. Each other method is given as a Gaussian distribution with mean and standard deviation as is presented in Table 2.

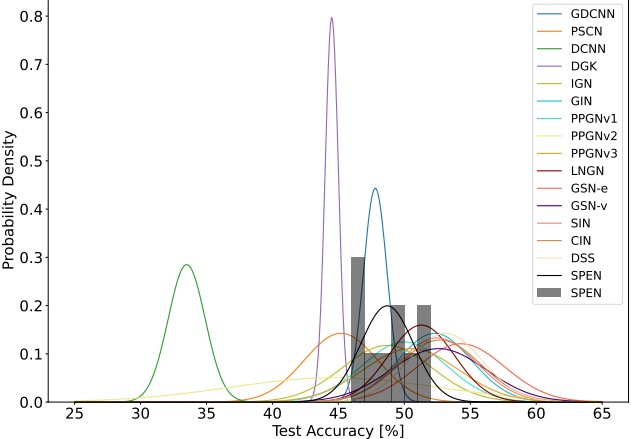

Figure 14: Comparison between our SPEN method and other methods on the IMDB-M dataset. Results for the SPEN method are also presented as a histogram of the 10-fold runs. Each other method is given as a Gaussian distribution with mean and standard deviation as is presented in Table 2.

Table 5: Statistical significant analysis of results on the MUTAG dataset given as p-values. The null hypothesis is that each model produces the same accuracy. A p-value of less than 0.05 is required to reject this null hypothesis and thus conclude that two models produce different accuracy's.

| | GDCNN | PSCN | DGK | CCN | IGN | GIN | PPGN v1 | PPGN v2 | PPGN v3 | LNGN | GSN-e | GSN-v | CIN | DSS | SPEN |
|---|---|---|---|---|---|---|---|---|---|---|---|---|---|---|---|
| GDCNN | | 0.05 | 0.13 | 0.03 | 0.66 | 0.08 | 0.13 | 0.23 | 0.20 | 0.00 | 0.08 | 0.03 | 0.01 | 0.01 | 0.01 |
| PSCN | 0.05 | | 0.34 | 0.35 | 0.26 | 0.86 | 0.63 | 0.97 | 0.89 | 0.79 | 0.57 | 0.26 | 0.14 | 0.25 | 0.10 |
| DGK | 0.13 | 0.34 | | 0.11 | 0.42 | 0.33 | 0.31 | 0.56 | 0.47 | 0.06 | 0.23 | 0.08 | 0.03 | 0.04 | 0.02 |
| CCN | 0.03 | 0.35 | 0.11 | | 0.12 | 0.46 | 0.76 | 0.42 | 0.53 | 0.37 | 0.76 | 0.86 | 0.72 | 0.97 | 0.59 |
| IGN | 0.66 | 0.26 | 0.42 | 0.12 | | 0.24 | 0.20 | 0.31 | 0.27 | 0.22 | 0.18 | 0.10 | 0.08 | 0.11 | 0.06 |
| GIN | 0.08 | 0.86 | 0.33 | 0.46 | 0.24 | | 0.74 | 0.87 | 1.00 | 1.00 | 0.69 | 0.36 | 0.22 | 0.38 | 0.17 |
| PPGN v1 | 0.13 | 0.63 | 0.31 | 0.76 | 0.20 | 0.74 | | 0.66 | 0.77 | 0.70 | 0.98 | 0.65 | 0.52 | 0.76 | 0.43 |
| PPGN v2 | 0.23 | 0.97 | 0.56 | 0.42 | 0.31 | 0.87 | 0.66 | | 0.89 | 0.84 | 0.62 | 0.34 | 0.23 | 0.37 | 0.18 |
| PPGN v3 | 0.20 | 0.89 | 0.47 | 0.53 | 0.27 | 1.00 | 0.77 | 0.89 | | 1.00 | 0.74 | 0.43 | 0.32 | 0.49 | 0.25 |
| LNGN | 0.00 | 0.79 | 0.06 | 0.37 | 0.22 | 1.00 | 0.70 | 0.84 | 1.00 | | 0.63 | 0.28 | 0.13 | 0.22 | 0.09 |
| GSN-e | 0.08 | 0.57 | 0.23 | 0.76 | 0.18 | 0.69 | 0.98 | 0.62 | 0.74 | 0.63 | | 0.64 | 0.50 | 0.75 | 0.40 |
| GSN-v | 0.03 | 0.26 | 0.08 | 0.86 | 0.10 | 0.36 | 0.65 | 0.34 | 0.43 | 0.28 | 0.64 | | 0.87 | 0.81 | 0.73 |
| CIN | 0.01 | 0.14 | 0.03 | 0.72 | 0.08 | 0.22 | 0.52 | 0.23 | 0.32 | 0.13 | 0.50 | 0.87 | | 0.63 | 0.83 |
| DSS | 0.01 | 0.25 | 0.04 | 0.97 | 0.11 | 0.38 | 0.76 | 0.37 | 0.49 | 0.22 | 0.75 | 0.81 | 0.63 | | 0.49 |
| SPEN | 0.01 | 0.10 | 0.02 | 0.59 | 0.06 | 0.17 | 0.43 | 0.18 | 0.25 | 0.09 | 0.40 | 0.73 | 0.83 | 0.49 | |

Table 6: Statistical significant analysis of results on the PTC dataset given as p-values. The null hypothesis is that each model produces the same accuracy. A p-value of less than 0.05 is required to reject this null hypothesis and thus conclude that two models produce different accuracy's.

| | GDCNN | PSCN | DGK | CCN | IGN | GIN | PPGN v1 | PPGN v2 | PPGN v3 | LNGN | GSN-e | GSN-v | CIN | DSS | SPEN |
|---|---|---|---|---|---|---|---|---|---|---|---|---|---|---|---|
| GDCNN | | 0.08 | 0.21 | 0.00 | 0.97 | 0.03 | 0.01 | 0.03 | 0.09 | 0.00 | 0.00 | 0.00 | 0.00 | 0.00 | 0.00 |
| PSCN | 0.08 | | 0.29 | 0.01 | 0.20 | 0.43 | 0.17 | 0.43 | 0.84 | 0.04 | 0.06 | 0.06 | 0.03 | 0.04 | 0.02 |
| DGK | 0.21 | 0.29 | | 0.00 | 0.51 | 0.08 | 0.02 | 0.09 | 0.26 | 0.00 | 0.01 | 0.00 | 0.00 | 0.00 | 0.01 |
| CCN | 0.00 | 0.01 | 0.00 | | 0.00 | 0.07 | 0.16 | 0.09 | 0.02 | 0.13 | 0.46 | 0.28 | 0.41 | 0.39 | 0.86 |
| IGN | 0.97 | 0.20 | 0.51 | 0.00 | | 0.07 | 0.02 | 0.07 | 0.17 | 0.00 | 0.01 | 0.01 | 0.00 | 0.00 | 0.00 |
| GIN | 0.03 | 0.43 | 0.08 | 0.07 | 0.07 | | 0.60 | 0.98 | 0.59 | 0.36 | 0.27 | 0.34 | 0.22 | 0.26 | 0.10 |
| PPGN v1 | 0.01 | 0.17 | 0.02 | 0.16 | 0.02 | 0.60 | | 0.64 | 0.29 | 0.78 | 0.52 | 0.67 | 0.47 | 0.53 | 0.19 |
| PPGN v2 | 0.03 | 0.43 | 0.09 | 0.09 | 0.07 | 0.98 | 0.64 | | 0.59 | 0.41 | 0.30 | 0.38 | 0.25 | 0.30 | 0.11 |
| PPGN v3 | 0.09 | 0.84 | 0.26 | 0.02 | 0.17 | 0.59 | 0.29 | 0.59 | | 0.12 | 0.11 | 0.13 | 0.08 | 0.10 | 0.04 |
| LNGN | 0.00 | 0.04 | 0.00 | 0.13 | 0.00 | 0.36 | 0.78 | 0.41 | 0.12 | | 0.56 | 0.76 | 0.47 | 0.56 | 0.18 |
| GSN-e | 0.00 | 0.06 | 0.01 | 0.46 | 0.01 | 0.27 | 0.52 | 0.30 | 0.11 | 0.56 | | 0.79 | 1.00 | 0.95 | 0.43 |
| GSN-v | 0.00 | 0.06 | 0.00 | 0.28 | 0.01 | 0.34 | 0.67 | 0.38 | 0.13 | 0.76 | 0.79 | | 0.76 | 0.82 | 0.29 |
| CIN | 0.00 | 0.03 | 0.00 | 0.41 | 0.00 | 0.22 | 0.47 | 0.25 | 0.08 | 0.47 | 1.00 | 0.76 | | 0.94 | 0.40 |
| DSS | 0.00 | 0.04 | 0.00 | 0.39 | 0.00 | 0.26 | 0.53 | 0.30 | 0.10 | 0.56 | 0.95 | 0.82 | 0.94 | | 0.38 |
| SPEN | 0.00 | 0.02 | 0.01 | 0.86 | 0.00 | 0.10 | 0.19 | 0.11 | 0.04 | 0.18 | 0.43 | 0.29 | 0.40 | 0.38 | |

Table 7: Statistical significant analysis of results on the PROTEINS dataset given as p-values. The null hypothesis is that each model produces the same accuracy. A p-value of less than 0.05 is required to reject this null hypothesis and thus conclude that two models produce different accuracy's.

| | GDCNN | PSCN | DCNN | DGK | IGN | GIN | PPGN v1 | PPGN v2 | PPGN v3 | LNGN | GSN-e | GSN-v | SIN | CIN | DSS | SPEN |
|---|---|---|---|---|---|---|---|---|---|---|---|---|---|---|---|---|
| GDCNN | | 0.56 | 0.00 | 0.55 | 0.55 | 0.47 | 0.29 | 0.59 | 0.52 | 0.00 | 0.51 | 0.59 | 0.39 | 0.31 | 0.48 | 0.52 |
| PSCN | 0.56 | | 0.00 | 0.41 | 0.42 | 0.33 | 0.21 | 0.44 | 0.40 | 0.00 | 0.38 | 0.82 | 0.28 | 0.22 | 0.35 | 0.88 |
| DCNN | 0.00 | 0.00 | | 0.00 | 0.00 | 0.00 | 0.00 | 0.00 | 0.00 | 0.00 | 0.00 | 0.00 | 0.00 | 0.00 | 0.00 | 0.00 |
| DGK | 0.55 | 0.41 | 0.00 | | 0.62 | 0.59 | 0.34 | 0.67 | 0.59 | 0.00 | 0.58 | 0.51 | 0.48 | 0.37 | 0.55 | 0.40 |
| IGN | 0.55 | 0.42 | 0.00 | 0.62 | | 0.84 | 0.80 | 0.93 | 0.97 | 0.02 | 1.00 | 0.41 | 0.96 | 0.86 | 1.00 | 0.39 |
| GIN | 0.47 | 0.33 | 0.00 | 0.59 | 0.84 | | 0.57 | 0.91 | 0.80 | 0.00 | 0.83 | 0.39 | 0.83 | 0.63 | 0.82 | 0.31 |
| PPGN v1 | 0.29 | 0.21 | 0.00 | 0.34 | 0.80 | 0.57 | | 0.72 | 0.83 | 0.00 | 0.79 | 0.25 | 0.71 | 0.92 | 0.78 | 0.20 |
| PPGN v2 | 0.59 | 0.44 | 0.00 | 0.67 | 0.93 | 0.91 | 0.72 | | 0.90 | 0.02 | 0.93 | 0.43 | 0.96 | 0.78 | 0.93 | 0.41 |
| PPGN v3 | 0.52 | 0.40 | 0.00 | 0.59 | 0.97 | 0.80 | 0.83 | 0.90 | | 0.02 | 0.97 | 0.39 | 0.92 | 0.89 | 0.97 | 0.37 |
| LNGN | 0.00 | 0.00 | 0.00 | 0.00 | 0.02 | 0.00 | 0.00 | 0.02 | 0.02 | | 0.01 | 0.10 | 0.00 | 0.00 | 0.01 | 0.01 |
| GSN-e | 0.51 | 0.38 | 0.00 | 0.58 | 1.00 | 0.83 | 0.79 | 0.93 | 0.97 | 0.01 | | 0.38 | 0.96 | 0.85 | 1.00 | 0.35 |
| GSN-v | 0.59 | 0.82 | 0.00 | 0.51 | 0.41 | 0.39 | 0.25 | 0.43 | 0.39 | 0.10 | 0.38 | | 0.34 | 0.27 | 0.36 | 0.92 |
| SIN | 0.39 | 0.28 | 0.00 | 0.48 | 0.96 | 0.83 | 0.71 | 0.96 | 0.92 | 0.00 | 0.96 | 0.34 | | 0.78 | 0.96 | 0.26 |
| CIN | 0.31 | 0.22 | 0.00 | 0.37 | 0.86 | 0.63 | 0.92 | 0.78 | 0.89 | 0.00 | 0.85 | 0.27 | 0.78 | | 0.84 | 0.21 |
| DSS | 0.48 | 0.35 | 0.00 | 0.55 | 1.00 | 0.82 | 0.78 | 0.93 | 0.97 | 0.01 | 1.00 | 0.36 | 0.96 | 0.84 | | 0.32 |
| SPEN | 0.52 | 0.88 | 0.00 | 0.40 | 0.39 | 0.31 | 0.20 | 0.41 | 0.37 | 0.01 | 0.35 | 0.92 | 0.26 | 0.21 | 0.32 | |

Table 8: Statistical significant analysis of results on the NCI1 dataset given as p-values. The null hypothesis is that each model produces the same accuracy. A p-value of less than 0.05 is required to reject this null hypothesis and thus conclude that two models produce different accuracy's.

| | GDCNN | PSCN | DCNN | DGK | CCN | IGN | GIN | PPGN v1 | PPGN v2 | PPGN v3 | LNGN | GSN-e | GSN-v | SIN | CIN | DSS | SPEN |
|---|---|---|---|---|---|---|---|---|---|---|---|---|---|---|---|---|---|
| GDCNN | | 0.01 | 0.00 | 0.00 | 0.18 | 0.91 | 0.00 | 0.00 | 0.00 | 0.00 | 0.00 | 0.00 | 0.00 | 0.00 | 0.00 | 0.00 | 0.00 |
| PSCN | 0.01 | | 0.00 | 0.00 | 1.00 | 0.07 | 0.00 | 0.00 | 0.00 | 0.00 | 0.00 | 0.00 | 0.00 | 0.00 | 0.00 | 0.00 | 0.00 |
| DCNN | 0.00 | 0.00 | | 0.00 | 0.00 | 0.00 | 0.00 | 0.00 | 0.00 | 0.00 | 0.00 | 0.00 | 0.00 | 0.00 | 0.00 | 0.00 | 0.00 |
| DGK | 0.00 | 0.00 | 0.00 | | 0.01 | 0.00 | 0.00 | 0.00 | 0.22 | 0.29 | 0.00 | 0.07 | 0.07 | 0.05 | 0.04 | 0.00 | 0.00 |
| CCN | 0.18 | 1.00 | 0.00 | 0.01 | | 0.22 | 0.00 | 0.00 | 0.00 | 0.01 | 0.00 | 0.00 | 0.00 | 0.00 | 0.00 | 0.00 | 0.00 |
| IGN | 0.91 | 0.07 | 0.00 | 0.00 | 0.22 | | 0.00 | 0.00 | 0.00 | 0.00 | 0.00 | 0.00 | 0.00 | 0.00 | 0.00 | 0.00 | 0.00 |
| GIN | 0.00 | 0.00 | 0.00 | 0.00 | 0.00 | 0.00 | | 0.45 | 0.10 | 0.05 | 1.00 | 0.64 | 0.64 | 0.93 | 0.55 | 0.23 | 0.18 |
| PPGN v1 | 0.00 | 0.00 | 0.00 | 0.00 | 0.00 | 0.00 | 0.45 | | 0.02 | 0.01 | 0.39 | 0.86 | 0.86 | 0.73 | 0.78 | 0.55 | 0.41 |
| PPGN v2 | 0.00 | 0.00 | 0.00 | 0.22 | 0.00 | 0.00 | 0.10 | 0.02 | | 0.83 | 0.08 | 0.20 | 0.20 | 0.22 | 0.14 | 0.01 | 0.01 |
| PPGN v3 | 0.00 | 0.00 | 0.00 | 0.29 | 0.01 | 0.00 | 0.05 | 0.01 | 0.83 | | 0.04 | 0.17 | 0.17 | 0.17 | 0.11 | 0.00 | 0.00 |
| LNGN | 0.00 | 0.00 | 0.00 | 0.00 | 0.00 | 0.00 | 1.00 | 0.39 | 0.08 | 0.04 | | 0.64 | 0.64 | 0.93 | 0.54 | 0.17 | 0.14 |
| GSN-e | 0.00 | 0.00 | 0.00 | 0.07 | 0.00 | 0.00 | 0.64 | 0.86 | 0.20 | 0.17 | 0.64 | | 1.00 | 0.72 | 0.96 | 1.00 | 0.91 |
| GSN-v | 0.00 | 0.00 | 0.00 | 0.07 | 0.00 | 0.00 | 0.64 | 0.86 | 0.20 | 0.17 | 0.64 | 1.00 | | 0.72 | 0.96 | 1.00 | 0.91 |
| SIN | 0.00 | 0.00 | 0.00 | 0.05 | 0.00 | 0.00 | 0.93 | 0.73 | 0.22 | 0.17 | 0.93 | 0.72 | 0.72 | | 0.65 | 0.55 | 0.46 |
| CIN | 0.00 | 0.00 | 0.00 | 0.04 | 0.00 | 0.00 | 0.55 | 0.78 | 0.14 | 0.11 | 0.54 | 0.96 | 0.96 | 0.65 | | 0.94 | 0.95 |
| DSS | 0.00 | 0.00 | 0.00 | 0.00 | 0.00 | 0.00 | 0.23 | 0.55 | 0.01 | 0.00 | 0.17 | 1.00 | 1.00 | 0.55 | 0.94 | | 0.74 |
| SPEN | 0.00 | 0.00 | 0.00 | 0.00 | 0.00 | 0.00 | 0.18 | 0.41 | 0.01 | 0.00 | 0.14 | 0.91 | 0.91 | 0.46 | 0.95 | 0.74 | |

Table 9: Statistical significant analysis of results on the NCI109 dataset given as p-values. The null hypothesis is that each model produces the same accuracy. A p-value of less than 0.05 is required to reject this null hypothesis and thus conclude that two models produce different accuracy's.

| | DGK | CCN | IGN | PPGN v1 | PPGN v2 | PPGN v3 | LNGN | CIN | DSS | SPEN |
|---|---|---|---|---|---|---|---|---|---|---|
| DGK | | 0.00 | 0.00 | 0.03 | 0.01 | 0.00 | 0.00 | 0.00 | 0.00 | 0.00 |
| CCN | 0.00 | | 0.04 | 0.00 | 0.00 | 0.00 | 0.00 | 0.00 | 0.00 | 0.00 |
| IGN | 0.00 | 0.04 | | 0.00 | 0.00 | 0.00 | 0.00 | 0.00 | 0.00 | 0.00 |
| PPGN v1 | 0.03 | 0.00 | 0.00 | | 1.00 | 0.60 | 0.17 | 0.01 | 0.38 | 0.04 |
| PPGN v2 | 0.01 | 0.00 | 0.00 | 1.00 | | 0.52 | 0.12 | 0.00 | 0.30 | 0.01 |
| PPGN v3 | 0.00 | 0.00 | 0.00 | 0.60 | 0.52 | | 0.30 | 0.02 | 0.66 | 0.05 |
| LNGN | 0.00 | 0.00 | 0.00 | 0.17 | 0.12 | 0.30 | | 0.22 | 0.53 | 0.58 |
| CIN | 0.00 | 0.00 | 0.00 | 0.01 | 0.00 | 0.02 | 0.22 | | 0.05 | 0.36 |
| DSS | 0.00 | 0.00 | 0.00 | 0.38 | 0.30 | 0.66 | 0.53 | 0.05 | | 0.17 |
| SPEN | 0.00 | 0.00 | 0.00 | 0.04 | 0.01 | 0.05 | 0.58 | 0.36 | 0.17 | |

Table 10: Statistical significant analysis of results on the IMDB-B dataset given as p-values. The null hypothesis is that each model produces the same accuracy. A p-value of less than 0.05 is required to reject this null hypothesis and thus conclude that two models produce different accuracy's.

| | GDCNN | PSCN | DCNN | DGK | IGN | GIN | PPGN v1 | PPGN v2 | PPGN v3 | LNGN | GSN-e | GSN-v | SIN | CIN | DSS | SPEN |
|---|---|---|---|---|---|---|---|---|---|---|---|---|---|---|---|---|
| GDCNN | | 0.23 | 0.00 | 0.00 | 0.28 | 0.01 | 0.13 | 0.15 | 0.14 | 0.00 | 0.00 | 0.00 | 0.00 | 0.00 | 0.00 | 0.00 |
| PSCN | 0.23 | | 0.00 | 0.00 | 0.61 | 0.04 | 0.37 | 0.45 | 0.33 | 0.00 | 0.00 | 0.00 | 0.00 | 0.00 | 0.00 | 0.00 |
| DCNN | 0.00 | 0.00 | | 0.00 | 0.00 | 0.00 | 0.00 | 0.00 | 0.00 | 0.00 | 0.00 | 0.00 | 0.00 | 0.00 | 0.00 | 0.00 |
| DGK | 0.00 | 0.00 | 0.00 | | 0.02 | 0.00 | 0.01 | 0.00 | 0.01 | 0.00 | 0.00 | 0.00 | 0.00 | 0.00 | 0.00 | 0.00 |
| IGN | 0.28 | 0.61 | 0.00 | 0.02 | | 0.21 | 0.80 | 0.93 | 0.70 | 0.16 | 0.01 | 0.02 | 0.09 | 0.11 | 0.06 | 0.13 |
| GIN | 0.01 | 0.04 | 0.00 | 0.00 | 0.21 | | 0.28 | 0.19 | 0.40 | 0.87 | 0.18 | 0.35 | 0.80 | 0.80 | 0.55 | 0.96 |
| PPGN v1 | 0.13 | 0.37 | 0.00 | 0.01 | 0.80 | 0.28 | | 0.85 | 0.87 | 0.21 | 0.01 | 0.03 | 0.13 | 0.14 | 0.07 | 0.18 |
| PPGN v2 | 0.15 | 0.45 | 0.00 | 0.00 | 0.93 | 0.19 | 0.85 | | 0.73 | 0.11 | 0.00 | 0.01 | 0.06 | 0.07 | 0.03 | 0.09 |
| PPGN v3 | 0.14 | 0.33 | 0.00 | 0.01 | 0.70 | 0.40 | 0.87 | 0.73 | | 0.37 | 0.04 | 0.08 | 0.23 | 0.25 | 0.15 | 0.31 |
| LNGN | 0.00 | 0.00 | 0.00 | 0.00 | 0.16 | 0.87 | 0.21 | 0.11 | 0.37 | | 0.03 | 0.04 | 0.51 | 0.56 | 0.27 | 0.74 |
| GSN-e | 0.00 | 0.00 | 0.00 | 0.00 | 0.01 | 0.18 | 0.01 | 0.00 | 0.04 | 0.03 | | 0.43 | 0.15 | 0.18 | 0.34 | 0.09 |
| GSN-v | 0.00 | 0.00 | 0.00 | 0.00 | 0.02 | 0.35 | 0.03 | 0.01 | 0.08 | 0.04 | 0.43 | | 0.33 | 0.38 | 0.71 | 0.19 |
| SIN | 0.00 | 0.00 | 0.00 | 0.00 | 0.09 | 0.80 | 0.13 | 0.06 | 0.23 | 0.51 | 0.15 | 0.33 | | 1.00 | 0.65 | 0.78 |
| CIN | 0.00 | 0.00 | 0.00 | 0.00 | 0.11 | 0.80 | 0.14 | 0.07 | 0.25 | 0.56 | 0.18 | 0.38 | 1.00 | | 0.67 | 0.80 |
| DSS | 0.00 | 0.00 | 0.00 | 0.00 | 0.06 | 0.55 | 0.07 | 0.03 | 0.15 | 0.27 | 0.34 | 0.71 | 0.65 | 0.67 | | 0.47 |
| SPEN | 0.00 | 0.00 | 0.00 | 0.00 | 0.13 | 0.96 | 0.18 | 0.09 | 0.31 | 0.74 | 0.09 | 0.19 | 0.78 | 0.80 | 0.47 | |

Table 11: Statistical significant analysis of results on the IMDB-M dataset given as p-values. The null hypothesis is that each model produces the same accuracy. A p-value of less than 0.05 is required to reject this null hypothesis and thus conclude that two models produce different accuracy's.

| | GDCNN | PSCN | DCNN | DGK | IGN | GIN | PPGNv1 | PPGNv2 | PPGNv3 | LNGN | GSN-e | GSN-v | SIN | CIN | DSS | SPEN |
|---|---|---|---|---|---|---|---|---|---|---|---|---|---|---|---|---|
| GDCNN | | 0.02 | 0.00 | 0.00 | 0.44 | 0.00 | 0.06 | 0.25 | 0.04 | 0.00 | 0.00 | 0.00 | 0.00 | 0.00 | 0.00 | 0.22 |
| PSCN | 0.02 | | 0.00 | 0.46 | 0.02 | 0.00 | 0.00 | 0.85 | 0.00 | 0.00 | 0.00 | 0.00 | 0.00 | 0.00 | 0.00 | 0.01 |
| DCNN | 0.00 | 0.00 | | 0.00 | 0.00 | 0.00 | 0.00 | 0.00 | 0.00 | 0.00 | 0.00 | 0.00 | 0.00 | 0.00 | 0.00 | 0.00 |
| DGK | 0.00 | 0.46 | 0.00 | | 0.00 | 0.00 | 0.00 | 0.94 | 0.00 | 0.00 | 0.00 | 0.00 | 0.00 | 0.00 | 0.00 | 0.00 |
| IGN | 0.44 | 0.02 | 0.00 | 0.00 | | 0.02 | 0.39 | 0.17 | 0.27 | 0.07 | 0.00 | 0.02 | 0.02 | 0.01 | 0.01 | 1.00 |
| GIN | 0.00 | 0.00 | 0.00 | 0.00 | 0.02 | | 0.10 | 0.02 | 0.23 | 0.41 | 0.16 | 0.84 | 0.88 | 0.77 | 0.53 | 0.00 |
| PPGNv1 | 0.06 | 0.00 | 0.00 | 0.00 | 0.39 | 0.10 | | 0.07 | 0.75 | 0.33 | 0.01 | 0.11 | 0.09 | 0.07 | 0.03 | 0.29 |
| PPGNv2 | 0.25 | 0.85 | 0.00 | 0.94 | 0.17 | 0.02 | 0.07 | | 0.06 | 0.03 | 0.00 | 0.01 | 0.01 | 0.01 | 0.01 | 0.15 |
| PPGNv3 | 0.04 | 0.00 | 0.00 | 0.00 | 0.27 | 0.23 | 0.75 | 0.06 | | 0.57 | 0.02 | 0.21 | 0.19 | 0.16 | 0.09 | 0.19 |
| LNGN | 0.00 | 0.00 | 0.00 | 0.00 | 0.07 | 0.41 | 0.33 | 0.03 | 0.57 | | 0.04 | 0.36 | 0.34 | 0.28 | 0.15 | 0.02 |
| GSN-e | 0.00 | 0.00 | 0.00 | 0.00 | 0.00 | 0.16 | 0.01 | 0.00 | 0.02 | 0.04 | | 0.29 | 0.22 | 0.28 | 0.39 | 0.00 |
| GSN-v | 0.00 | 0.00 | 0.00 | 0.00 | 0.02 | 0.84 | 0.11 | 0.01 | 0.21 | 0.36 | 0.29 | | 0.95 | 0.95 | 0.73 | 0.01 |
| SIN | 0.00 | 0.00 | 0.00 | 0.00 | 0.02 | 0.88 | 0.09 | 0.01 | 0.19 | 0.34 | 0.22 | 0.95 | | 0.89 | 0.65 | 0.00 |
| CIN | 0.00 | 0.00 | 0.00 | 0.00 | 0.01 | 0.77 | 0.07 | 0.01 | 0.16 | 0.28 | 0.28 | 0.95 | 0.89 | | 0.77 | 0.00 |
| DSS | 0.00 | 0.00 | 0.00 | 0.00 | 0.01 | 0.53 | 0.03 | 0.01 | 0.09 | 0.15 | 0.39 | 0.73 | 0.65 | 0.77 | | 0.00 |
| SPEN | 0.22 | 0.01 | 0.00 | 0.00 | 1.00 | 0.00 | 0.29 | 0.15 | 0.19 | 0.02 | 0.00 | 0.01 | 0.00 | 0.00 | 0.00 | |

