# OpenReview forum: "Subgraph Permutation Equivariant Networks"
_TMLR — Rejected by TMLR_

### Review · Reviewer_bX4D · 2022-08-15

**Summary Of Contributions:**

This paper proposes to extract sub-graph from the original one and explore automorphism in these sub-graphs. The overall GNN architecture is then obtained by stacking multiple layers, which are comprised of automorphism equivalent mapping functions. Finally, theoretical analysis shows the proposed architecture is strictly better than sub-graph MPNN.

**Broader Impact Concerns:**

None.

**Requested Changes:**

Thanks for authors' efforts in addressing comments raised in last round's review, but they are still not sufficient addressed. Please see weakness above.

**Strengths And Weaknesses:**

Strength.
- The idea is overall interesting and well-motivated.
- Theoretical analysis is valid and comprehensive.

Weakness. Still, discussion with related works and experimental evaluations are the main concern.
- As mentioned last time, considering automorphism in GNN is not first introduced by the authors. It is better add more discussion with those existing works in the main text. Currently, it is briefly done above Section 3, but lots of discussions are in Appendix A.3. I still think existing works should be more emphasized in the main text.

- Why not methods in Appendix A.3 compared in Table 2, especially after so much discussion in Appendix A.3? ESAN / Autobahn / GRAPE. I still think this part is important to understand position of this work in the literature.

- Lack ablation study of the proposed method. For example, how the depth of the architecture will influence the performance / how the k, i.e., k-hop connectivities, will affect the performance. what if subgraph linear maps are not used (i.e., no need to ensure automorphism, just plain activation functions or MLP)?

If authors feel the space in the main text is not enough, perhaps, move some definitions / Figure 5 to the appendix.

---

> ### Author Response · Authors · 2022-08-15
> **Response to Reviewer bX4D**
>
> - As mentioned last time, considering automorphism in GNN is not first introduced by the authors. It is better add more discussion with those existing works in the main text. Currently, it is briefly done above Section 3, but lots of discussions are in Appendix A.3. I still think existing works should be more emphasized in the main text.
>
> We have 3 pages of discussion on previous methods in Appendix A.3. and half a page of discussion on previous methods in the main text. Many other papers published in this area also have a half a page discussion on previous methods (or less) [1,2,3,4,5,6]. Further, most of these papers do not provide the connection to previous work in the Appendix like we do. This is not supposed to be a survey paper and we believe 3 pages of discussion of previous work (a quarter of the paper) is too much. We could move a figure to the appendix and bring some additional discussion into the main text or alternatively extend the paper beyond 12 pages if all reviewers agree this would be the correct approach. Can the reviewer be clearer as to what precisely they feel is missing currently in the main text which stops this paper from being ready for publication, as it is?
>
>
> - Why not methods in Appendix A.3 compared in Table 2, especially after so much discussion in Appendix A.3? ESAN / Autobahn / GRAPE. I still think this part is important to understand position of this work in the literature.
>
> ESAN is already compared to - we used the name DSS (EGO) as was done in their paper.
>
> Autobahn does not test on the TU Datasets benchmarks, which are used by most papers. Therefore we cannot compare the experimental results, and this was explained in the previous review phase.
>
> GRAPE also does not test on the TU Datasets benchmarks, which are used by most papers. Therefore we cannot compare the experimental results. GRAPE used benchmarks such as Cora, which we believe are widely acknowledged to not be challenging enough moving forward and hence we didn’t use them.
>
>
> - Lack of ablation study of the proposed method. For example, how the depth of the architecture will influence the performance / how the k, i.e., k-hop connectivities, will affect the performance. what if subgraph linear maps are not used (i.e., no need to ensure automorphism, just plain activation functions or MLP)?
>
> The majority of other methods in this area of research do not conduct an ablation study of their proposed methods  [1,2,3,4,5,6] (published in competitive venues suchs NeurIPS,ICML,ICLR)  as it is computationally expensive to run on these benchmarks. There are 7 datasets of which 10-fold cross validation is used on each, therefore requiring the training of 70 models. We have demonstrated that our method is competitive with the SOTA methods and improved the experimentation section in line with the requests from the AC following the previous review cycle (relating to the statistical analysis, where again we have now gone beyond the state of the art in terms of statistical analysis in this area).
>
>
> [1] Maron, H., Ben-Hamu, H., Shamir, N. and Lipman, Y., 2018. Invariant and equivariant graph networks. arXiv preprint arXiv:1812.09902.
>
> [2] Bevilacqua, B., Frasca, F., Lim, D., Srinivasan, B., Cai, C., Balamurugan, G., Bronstein, M.M. and Maron, H., 2021. Equivariant subgraph aggregation networks. arXiv preprint arXiv:2110.02910.
>
> [3] Bodnar, C., Frasca, F., Otter, N., Wang, Y., Lio, P., Montufar, G.F. and Bronstein, M., 2021. Weisfeiler and Lehman go cellular: CW networks. Advances in Neural Information Processing Systems, 34, pp.2625-2640.
>
> [4] Bodnar, C., Frasca, F., Wang, Y., Otter, N., Montufar, G.F., Lio, P. and Bronstein, M., 2021, July. Weisfeiler and lehman go topological: Message passing simplicial networks. In International Conference on Machine Learning (pp. 1026-1037). PMLR.
>
> [5] Bouritsas, G., Frasca, F., Zafeiriou, S.P. and Bronstein, M., 2022. Improving graph neural network expressivity via subgraph isomorphism counting. IEEE Transactions on Pattern Analysis and Machine Intelligence.
>
> [6] de Haan, P., Cohen, T.S. and Welling, M., 2020. Natural graph networks. Advances in Neural Information Processing Systems, 33, pp.3636-3646.

---

### Review · Reviewer_N3Bm · 2022-08-25

**Summary Of Contributions:**

The paper proposes equivariant networks that operate on a graph as a nested bag-of-bags of subgraphs. Subgraphs are constructed as k-hop ego networks of each node, and the resulting bag of graphs is further partitioned into bags of similar size. A separate equivariant network is applied to each bag of subgraphs of similar size, and the resulting representations from different subgraphs containing a node are aggregated. A stack of such equivariant layers is used for learning on graphs. The paper presents theoretical analyses on the distinguishing power of this approach based on the k-WL test and provides experimental evidence on a graph classification benchmark.




**Requested Changes:**

N/A.

**Strengths And Weaknesses:**

Strength:
- Experiments showing the performance in terms of scalability are supportive
- Both theoretical and experimental evidence in support of the proposed method

Weakness:
- The main idea is quite incremental
- Discrepancy between theoretical presentation and experimental results

While I welcome changes since the first submission of the paper, the main problem in my initial review persists: unfortunately, I still don't see a clear motivation for separating subgraphs into groups if the symmetry group assumed for all these groups is the *symmetric group*. This is because we can apply the same network to symmetric groups of different sizes. The rebuttal argued that simply because we can apply the same network doesn't mean we should. On the contrary, I think the natural choice here is to share parameters unless there is a clear explanation as to why this is not a good idea. This problem becomes particularly important because of the closeness to several existing works and because the paper's technical presentation assumes different automorphism groups for subgraphs, while experiments use similar symmetric groups.

---

> ### Author Response · Authors · 2022-08-26
> **Response to Reviewer N3Bm**
>
> * unfortunately, I still don't see a clear motivation for separating subgraphs into groups if the symmetry group assumed for all these groups is the symmetric group. This is because we can apply the same network to symmetric groups of different sizes. The rebuttal argued that simply because we can apply the same network doesn't mean we should. On the contrary, I think the natural choice here is to share parameters unless there is a clear explanation as to why this is not a good idea. This problem becomes particularly important because of the closeness to several existing works and because the paper's technical presentation assumes different automorphism groups for subgraphs, while experiments use similar symmetric groups.
>
> There are two possible reasons for separating subgraphs into groups and therefore using the automorphism group and not sharing parameters. Firstly, subgraphs of different sizes can be distinctly different objects, and therefore treating them thusly is a useful inductive bias to build into a model. Secondly, permutation equivariance is a strong constraint to place upon the network, whilst automorphism equivariance is a weaker constraint. Permutation equivariance can be seen to be a strong constraint from comparing an unconstrained linear layer versus a permutation equivariant linear layer on graphs. In the case of a single feature dimension in the input and output, a permutation equivariant linear layer has 15 trainable parameters, whereas a unconstrained linear layer has $n^{4}$, for a graph with $n$ nodes, and in general $n^{4} >> 15$. On the other hand, an automorphism equivariant linear layer has, in our case where we use a permutation equivariant base GNN, $m \times 15$ trainable parameters, where $m$ is the number of automorphism groups. Except for the case where there is a single automorphism group, which is never in any of the datasets we considered, $m \times 15 > 15$, and therefore our automorphism equivariant model places a weaker constraint upon the network than permutation equivariant models, which gives the model more flexibility. We are happy to add this into the paper, can you let us know if it sufficiently addresses your concern around the motivation?
>
> Our technical presentation assumes different automorphism groups for subgraphs because this is the framework we are presenting and using in all experiments. The model used for the experiments is exactly the model presented throughout the paper, we do not use a different model for the experiments. To be extremely clear, we use the automorphism equivariant model for the experiments.
>
> * The main idea is quite incremental
>
> We have presented the key differences between our method and prior methods both in the main text and in the appendix. The key contributions are a novel way of encoding the graph as bags of sub-graphs, which leads to a novel choice of automorphism group; demonstrating improved scalability over the base GNN and improved expressivity.
>
> We have presented a theoretically coherent and novel contribution, and while we respect the reviewer’s right to believe that this is an incremental contribution, it cannot be denied that it is a contribution, and we would like to remind the reviewers and associate editor of the stated goals of the TMLR journal: https://openreview.net/group?id=TMLR
> “TMLR emphasizes technical correctness over subjective significance, in order to ensure we facilitate scientific discourses on topics that are deemed less significant by contemporaries but may be so in the future.”
>
> * Discrepancy between theoretical presentation and experimental results
>
> We assume the reviewer is referring to their perceived discrepancy: “the paper's technical presentation assumes different automorphism groups for subgraphs, while experiments use similar symmetric groups”. If this statement is saying that we present a technical method that operates on different automorphism groups on different bags of sub-graphs, but in the experiments we use the symmetric group and share weights between all sub-graphs, then this statement is wrong and the perceived discrepancy is wrong. We have made clear in the paper that our method is automorphism equivariant and each bag of sub-graphs is a different automorphism group. If we have misunderstood your discrepancy then can you please elaborate on what the discrepancy is?

---

> > ### Comment · Reviewer_N3Bm · 2022-09-20
> > **Clarifying the issue**
> >
> > Thanks for your response! To be provide further clarity, I'm giving a more concise version of my main issues with the current state of the paper. Please correct any misunderstanding on my part.
> >
> > **Presentation issue:** there is a repeated claim that the experiments use the automorphism group of subgraphs in the same bag ("To be extremely clear, we use the automorphism equivariant model for the experiments.") This seems misleading: the paper uses the symmetric group for all experiments according to Eq(2) of the appendix. Indeed using the automorphism group will be practically more challenging.
> >
> > **Technical issue:** this same choice makes the analysis and entire presentation and motivation of the model based on the "automorphism group" incompatible with the actual model used in the experiments.

---

> > > ### Author Response · Authors · 2022-09-21
> > > **Response to Reviewer N3Bm (1/2)**
> > >
> > > If we understand correctly there is essentially confusion about whether we are using the permutation group or the automorphism group. The reviewer believes we are using the permutation group and not considering the automorphism group, despite our repeated confirmation that we are in fact using the automorphism group.
> > >
> > > Firstly, we split the ego-net subgroups into bags such that each contains ego-net subgraphs of the same automorphism group. If we were simply using the permutation group this would be redundant and we may as well store all subgraphs in the same bag. We split into bags of subgraphs so that a different weight set can be used for each automorphism group, ensuring automorphism equivariance. Eq(2) of the appendix says that there is an automorphism between two graphs if there is some permutation action which makes them the same. If all subgraphs had the same structure then there would be one automorphism group, although this is not the case for any of the experiments we have considered and is unlikely to be the case in practice due to graphs being quite heterogeneous. Therefore, we have multiple automorphism groups. As such, we store subgraphs in bags according to their automorphism group and the model processes these in an automorphism equivariant manner, **and we use this automorphism equivariant model for the experiments**.
> > >
> > > We are absolutely confident that our model is processing bags of subgraphs in an automorphism equivariant model, which has a base update function that is permutation equivariant. As we provided in our first response, simply considering the number of learnable parameters in the model confirms that something is clearly different between our architecture and one that is just permutation equivariant. In a model that is permutation equivariant the number of trainable parameters in the weight matrix for a linear layer going from a graph feature space to graph feature space, each of feature dimension size 1, is $15$. On the other hand, our automorphism equivariant model has $m \times 15$, where $m$ is the number of automorphism groups. As previously stated in all of our examples $m > 1$. Therefore, there is a clear difference between our automorphism equivariant model and one which is not automorphism equivariant, and it is this automorphism equivariant model that we use for all the experiments.
> > >
> > > If there is simply confusion over the automorphism group we will provide a shortened version of the definition given in Introduction to Algebra by Peter J. Cameron:
> > >
> > > Automorphism Groups. Let $R$ be a ring. An **automorphism** of $R$ is an isomorphism $\theta : R \rightarrow R$; in other words, it is a permutation of $R$ which happens also to be a homomorphism.
> > >
> > > Let $\mathrm{Aut}(R)$ be the set of all automorphisms of $R$. Then $\mathrm{Aut}(R)$ is a group, the **automorphism group** of $R$.
> > >
> > > Remark. There is absolutely nothing special about rings here. If $\mathcal{X}$ is any class of mathematical objects for which we can formulate the notion of homomorphism (or isomorphism), then $\mathrm{Aut}(X)$ is a group for any $X \in \mathcal{X}$.
> > >
> > > We thank the reviewer for highlighting that *using the automorphism group will be practically more challenging* as this is the challenge that we have overcome here and this is exactly what we are doing. Perhaps if the reviewer can now see that we are using an automorphism equivariant architecture, they can see more value in our work.

---

> > > > ### Author Response · Authors · 2022-09-21
> > > > **Response to Reviewer N3Bm (2/2)**
> > > >
> > > > We have also provided a reduced version of the code to the main class of automorphism equivariant update functions below to highlight that we are using the automorphism groups, where ```weight_degrees``` holds the different automorphism groups. For this we are only showing the mapping between order 1 and 2 representations.
> > > >
> > > > ```
> > > > class eq_gconv(torch.nn.Module):
> > > >     def __init__(self, repin, repout, hid_dim1, hid_dim2, weight_degrees, subgraph_degrees, degree_mapping, k=1):
> > > >         super(eq_gconv, self).__init__()
> > > >         self.hid_dim1 = hid_dim1
> > > >         self.hid_dim2 = hid_dim2
> > > >         self.weight_degrees = weight_degrees
> > > >         self.subgraph_degrees = subgraph_degrees
> > > >         self.degree_mapping = degree_mapping
> > > >         self.k = k
> > > >         self.repin = repin
> > > >         self.repout = repout
> > > >
> > > >         if (repin==21) and (repout==21):
> > > >             eqblock_p2p2_list = []
> > > >             for i in self.weight_degrees:
> > > >                 eqblock_p2p2_list.append([f'{i}', eq.equi_2_to_2(self.hid_dim1, self.hid_dim2, 'cuda')])
> > > >             self.eqblock_p2p2 = torch.nn.ModuleDict(eqblock_p2p2_list)
> > > >             eqblock_p2p1_list = []
> > > >             for i in self.weight_degrees:
> > > >                 eqblock_p2p1_list.append([f'{i}', eq.equi_2_to_1(self.hid_dim1, self.hid_dim2, 'cuda')])
> > > >             self.eqblock_p2p1 = torch.nn.ModuleDict(eqblock_p2p1_list)
> > > >             eqblock_p1p2_list = []
> > > >             for i in self.weight_degrees:
> > > >                 eqblock_p1p2_list.append([f'{i}', eq.equi_1_to_2(self.hid_dim1, self.hid_dim2, 'cuda')])
> > > >             self.eqblock_p1p2 = torch.nn.ModuleDict(eqblock_p1p2_list)
> > > >             eqblock_p1p1_list = []
> > > >             for i in self.weight_degrees:
> > > >                 eqblock_p1p1_list.append([f'{i}', eq.equi_1_to_1(self.hid_dim1, self.hid_dim2, 'cuda')])
> > > >             self.eqblock_p1p1 = torch.nn.ModuleDict(eqblock_p1p1_list)
> > > >
> > > >             self.gnp1 = GraphNorm(2*self.hid_dim2)
> > > >             self.gnp2 = GraphNorm(2*self.hid_dim2)
> > > >
> > > >         self.elu = torch.nn.ELU()
> > > >
> > > >     def forward(self, data, T2, T1):
> > > >         T2_list, edge_indices_orig, T1_list, node_indices_orig, data, _ = get_subgraphs(data, T2, T1, self.subgraph_degrees, k=self.k)
> > > >
> > > >         T2_out = []
> > > >         if (self.repin==21) and (self.repout==21):
> > > >             for T2_in, T1_in in zip(T2_list, T1_list):
> > > >                 if (T2_in is not None) and (T1_in is not None):
> > > >                     deg_w = self.degree_mapping[f'{T2_in.shape[-1]}']
> > > >                     T2_out.append(torch.cat((self.eqblock_p2p2[deg_w](T2_in),self.eqblock_p1p2[deg_w](T1_in)), dim=1))
> > > >                     T1_out.append(torch.cat((self.eqblock_p2p1[deg_w](T2_in),self.eqblock_p1p1[deg_w](T1_in)), dim=1))
> > > >                 else:
> > > >                     T2_out.append(None)
> > > >                     T1_out.append(None)
> > > >
> > > >         if T2_out:
> > > >             edge_index_new, T2 = subgraphs_to_graph(data, T2_out, edge_indices_orig, self.subgraph_degrees)
> > > >             data.edge_index = edge_index_new
> > > >             T2 = self.elu(T2)
> > > >             T2 = self.gnp2(T2, data.batch[edge_index_new[0,:]])
> > > >         else:
> > > >             T2 = None
> > > >
> > > > ```

---

### Review · Reviewer_vbtW · 2022-09-08

**Summary Of Contributions:**

This paper presents an architecture for graph representation learning designed on the idea of extracting and processing "bags of subgraphs" separately in a way that is equivariant to the action of the automorphism groups of both the bags and the subgraphs.

The authors provide a fairly theoretical (as opposed to practical) description of the method under the light of group theory, proposing an architecture that acts on two levels

1) on the subgraphs, by processing them as graphs and sets
2) on the bags of subgraphs

The proposed architecture is shown to be strictly more expressive than conventional message-passing neural networks and to achieve results comparable to other higher-order GNNs while being more scalable than some methods.

**Requested Changes:**

- This sentence:

  >The natural numbers of the nodes are essential for representing the graphs in a computer, but hold no actual information about the underlying graph.

  is lifted almost verbatim from

  >de Haan, Pim, Taco S. Cohen, and Max Welling. "Natural graph networks." Advances in Neural Information Processing Systems 33 (2020): 3636-3646.

  I advise the authors to either re-phrase the passage or put it in quotes with a reference.

- Typo: "an natural question"


**Strengths And Weaknesses:**

# Strengths

- The paper looks solid and proposes a technique to overcome known issues in GNNs in a fairly efficient way.

- The paper is likely to be of interest to the research community that focuses on GNN expressivity, and to advance the conversation in that field.

- The discussion is thorough and, even though I have some concerns (see below), the authors are very transparent in the discussion of the limitations of their approach (e.g., the limited statistical significance of the results).


# Weaknesses

- Are the authors aware of this paper?
>Zhang, Muhan, and Pan Li. "Nested graph neural networks." Advances in Neural Information Processing Systems 34 (2021): 15734-15747.

  This looks like a relevant work that also considers rooted subgraphs as a way to overcome the limits of MPNNs in a computationally efficient way.
  Can the authors comment on the differences between the two works?

- The paper would enormously benefit by making the description of the architecture more "concrete". As it is, the chances that somebody would be able to implement the architecture from the description provided in Section 3.3 are slim.
In other words, given a graph as defined in Definition 3.2, how is it concretely acted upon by the various $f^S$ maps? It should be possible to summarize the actual implementation of the layer in an equation or algorithm, which I believe would help the readability/reproducibility of the paper.

- The results are underwhelming, since the authors can only conclude that their method is not significantly better or worse than the state of the art but on datasets that are known to be essentially useless for practical benchmarks.
This is still a helpful result since the proposed method is more efficient than other baselines. Still, it would have been more interesting to see results with at least some degree of significance on better benchmarks like OGB or Zinc.
Also, none of the reported results are close to state-of-the-art (from a quick check of the public leaderboards on paperswithcode.com), so the term should probably not be used here.
To be fair, this is a weakness of the paper but I realize that the benchmarking problem is widespread in the GNN community. It is likely that the authors would have been asked by some reviewers for these exact benchmarks, had they not reported them. I just advise the authors to also report meaningful experimental results, since we know that it is possible.

---

> ### Author Response · Authors · 2022-09-08
> **Response to Reviewer vbtW**
>
> * Are the authors aware of this paper?
> Zhang, Muhan, and Pan Li. "Nested graph neural networks." Advances in Neural Information Processing Systems 34 (2021): 15734-15747.
>
> We were not and will add in the citation and comparison to this work. A significant and key difference between this work and ours is that they do not consider automorphism groups, while we do.
>
> * The paper would enormously benefit by making the description of the architecture more "concrete". As it is, the chances that somebody would be able to implement the architecture from the description provided in Section 3.3 are slim. In other words, given a graph as defined in Definition 3.2, how is it concretely acted upon by the various $f^{S}$ maps? It should be possible to summarize the actual implementation of the layer in an equation or algorithm, which I believe would help the readability/reproducibility of the paper.
>
> We added four new figures to make more clear exactly how the architecture looks and how graphs are processed by it in the update based on prior comments. It should be possible based on the text and figure to implement the architecture, we also provide further details to assist with implementation in the appendix. Never-the-less, we have now added an algorithm of how the model works in Appendix A.6.2, please can you look at this and let us know if this addresses your concerns.
>
> * The results are underwhelming, since the authors can only conclude that their method is not significantly better or worse than the state of the art but on datasets that are known to be essentially useless for practical benchmarks. This is still a helpful result since the proposed method is more efficient than other baselines. Still, it would have been more interesting to see results with at least some degree of significance on better benchmarks like OGB or Zinc. Also, none of the reported results are close to state-of-the-art (from a quick check of the public leaderboards on paperswithcode.com), so the term should probably not be used here.
>
> We took the snapshot of algorithms which had wide applicability and had been tested on the wide range of benchmarks we chose for comparison, *and had appeared in peer-reviewed venues*. In a fast-moving field this will always have room for new entries, but we need to freeze the paper at some point, and if we continue adding changes there will always be new papers to add in. The TMLR guidelines highlight the need to judge for correctness what we actually did, not what else might be possible or preferred by reviewers. As an example, the UGformer algorithm appears top on paperswithcode.com on IMDB-B and appeared at WWW’22 in April (when we were submitting the initial version of this paper). It does significantly better on some of the methods, but is not compared to many of our benchmarks, and does worse on others (e.g. MUTAG), and also has not done the statistical significance analysis we did. Specific algorithms might do better on one or two benchmarks, but the point we are trying to show here is that we have presented a new approach which has conceptual benefits, and which also does roughly as well empirically overall as leading algorithms which were peer-reviewed at the time we did the research, and which have also been tested on a wide range of benchmarks.
>
> * Regarding the sentence issue and typo
>
> We will change these are the reviewer requests.

---

> > ### Comment · Reviewer_vbtW · 2022-09-13
> > **Reply**
> >
> > The authors have addressed my concerns, and I agree that the publication standards of TMLR are different than what is usually expected in terms of experiments.
> >
> > I still think that the paper could do a better job at explaining how the $f$ mappings are implemented (I could not find it in the algorithm mentioned by the authors), but this has little to do with technical correctness.
> >
> > Overall, I will recommend acceptance of the paper and I don't think there will be a need for submitting the paper elsewhere.